# A versatile soluble siglec scaffold for sensitive and quantitative detection of glycan ligands

Emily Rodrigues[1], Jaesoo Jung[1], Heajin Park[1], Caleb Loo[1], Sepideh Soukhtehzari[2], Elena N. Kitova[1], Fahima Mozaneh[1], Gour Daskhan[1], Edward N. Schmidt[1], Vivian Aghanya [1], Susmita Sarkar[1], Laura Streith [3], Chris D. St. Laurent[1], Linh Nguyen[1], Jean-Philippe Julien [4,5], Lori J. West[3,6,7], Karla C. Williams [2], John S. Klassen[1] & Matthew S. Macauley [1,3✉]

Sialic acid-binding immunoglobulin-type lectins (Siglecs) are immunomodulatory receptors that are regulated by their glycan ligands. The connections between Siglecs and human disease motivate improved methods to detect Siglec ligands. Here, we describe a new versatile set of Siglec-Fc proteins for glycan ligand detection. Enhanced sensitivity and selectivity are enabled through multimerization and avoiding Fc receptors, respectively. Using these Siglec-Fc proteins, Siglec ligands are systematically profiled on healthy and cancerous cells and tissues, revealing many unique patterns. Additional features enable the production of small, homogenous Siglec fragments and development of a quantitative ligand-binding mass spectrometry assay. Using this assay, the ligand specificities of several Siglecs are clarified. For CD33 (Siglec-3), we demonstrate that it recognizes both α2-3 and α2-6 sialosides in solution and on cells, which has implications for its link to Alzheimer's disease susceptibility. These soluble Siglecs reveal the abundance of their glycan ligands on host cells as self-associated molecular patterns.

[1] Department of Chemistry, University of Alberta, Edmonton, AB, Canada. [2] Department of Pharmaceutical Sciences, University of British Columbia, Vancouver, BC, Canada. [3] Department of Medical Microbiology and Immunology, University of Alberta, Edmonton, AB, Canada. [4] Departments of Biochemistry and Immunology, University of Toronto, Toronto, ON, Canada. [5] Program in Molecular Medicine, The Hospital for Sick Children Research Institute, Toronto, ON, Canada. [6] Department of Pediatrics, University of Alberta, Edmonton, AB, Canada. [7] Department of Surgery, University of Alberta, Edmonton, AB, Canada. ✉email: macauley@ualberta.ca

Sialic acid-binding immunoglobulin-type lectins (Siglecs) are a family of immunomodulatory receptors expressed on cells of the hematopoietic lineage[1]. As a rapidly evolving family, there are 15 Siglecs in humans, while mice only have 9 Siglecs. The majority of Siglecs are inhibitory, dampening immune cell signaling through immunoreceptor tyrosine-based inhibitory motifs (ITIMs) within their cytoplasmic domain[2]. Importantly, the immunomodulatory properties of Siglecs are regulated by interactions with sialic acid-containing glycoconjugates, which can be either on the same (*cis*) cell or opposing (*trans*) cell or particle.

Roles for Siglecs in health and disease have been rapidly expanding. Hypersialylation of cancer cells dampens immune cell responses and plays analogous, potentially overlapping, roles with immune checkpoints such as PD-1[3,4]. Hypersialylated tumors exploit Siglecs, including: Siglec-7 on NK cells[5], Siglec-9 on tumor-associated macrophages[6] and CD8+ T-cells[7], Siglec-10 on macrophages[8], and Siglec-15 on macrophages and/or tumors[9,10]. The exploitation of Siglecs by cancer may reflect the ability of Siglecs to aid in self/non-self-discrimination through recognition of sialic acid as a 'self-associated molecular pattern'[3].

To better understand Siglec-mediated regulation of immune cell function and develop more targeted approaches to block Siglec–ligand interactions, better methods are needed to probe the glycan ligands of Siglecs. A major determinant for Siglec specificity is the glycosidic linkage between the sialic acid and underlying α2-6- or α2-3-linked galactose, α2-6-linked *N*-acetylgalactosamine, or α2-8-linked sialic acid[1,11]. Additional underlying features also play a key role, such as sulfation and glycan type (N-glycan, O-glycan, or ganglioside). While much has been learned about the glycan ligands of Siglecs, there is still much to be explored.

One challenge in studying Siglec–glycan interactions is their low affinity. On the cell surface, multivalent presentation of both Siglecs and their glycans greatly enhance avidity. Avidity can also be leveraged to study Siglec–glycan interactions, with dimeric Siglec-Fc chimeras being a common approach to probe solid surfaces coated with glycans or cells[12]. The Fc domain has many advantages (dimerization, protein stability, and handle for protein purification/detection), but also has drawbacks, such as the requirement for strong acidic elution (Protein-A/G chromatography) and off-target binding to Fcγ receptors (FcγR)[13]. The use of αhuman IgG secondary antibody also has the drawback of binding deposited IgG on tissues. Accordingly, avoiding a secondary antibody would be desirable for detection of cellular Siglec ligands.

As knowledge about Siglec ligands predominantly comes from the use of Siglec-Fc chimeras, we were motived to improve this scaffold, as eliminating the aforementioned drawbacks would be highly desirable. Here, we present a new library of all 15 human Siglec-Fc constructs with improved capabilities for studying interactions with their sialic acid ligands on cells, tissues, and in solution. These re-engineered Siglec-Fc proteins allow for sensitive and selective sialic acid-dependent binding to cells and tissues, while monomeric, homogenous Siglec fragments derived from these constructs enables glycan interactions to be studied and quantified using a mass spectrometry-based approach.

## Results

**Design and purification of new Siglec-Fc constructs.** Aiming to generate soluble Siglecs that are compatible with a variety of assays, four features were incorporated: (1) a His$_6$-tag for purification that is directly C-terminal to the Fc domain; (2) a Strep-tag II for purification and detection; (3) mutations in the Fc domain (L234A, L235A, G237A, H268A, P238S, A330S, P331S)

to eliminate FcγR binding[14]; and (4) a TEV proteolytic cleavage site between the Siglec and Fc (Fig. 1a). Version 1 (V1), with the WT Fc, and version 2 (V2), with the mutated Fc, were initially created for Siglec-2 (CD22) (Supplementary Fig. 1a). Expression in stably-transfected CHO cells yielded good expression levels (~20 mg/L) that peaked 1-week post-confluence (Supplementary Fig. 1b), with supernatants from CHO cells expressing V1 and V2 of CD22 binding strongly to ST6Gal1-overexpressing CHO cells (Supplementary Fig. 1c). Therefore, we moved forward with generating the V2 construct for all 15 human Siglecs. The N-terminal three extracellular domains were used in all cases except Siglec-3 (CD33) and Siglec-15 because they only have two domains. A control construct for each Siglec was also created where the essential arginine required for sialic acid recognition[15] was mutated to alanine, except Siglec-12 that lacks this critical arginine[16]. All constructs were stably transfected in WT and Lec1 CHO cells, with expression levels ranging from 5 to 43 mg/L (Supplementary Table 1).

Using the V2 of Sig-6-Fc, we compared different protein purification strategies and found that double purification by Nickel column and Strep-Tactin column yielded highly purified protein (Supplementary Fig. 2a) and avoided strong acid elution for Protein-G chromatography. Size exclusion chromatography (SEC) was additionally carried out; using this three-step purification strategy, all the human Siglec-Fc chimeras were prepared (Fig. 1b). Highly purified protein functioned better in cell-binding experiments (Supplementary Fig. 2b), but activity decreased after several months of storage at 4 °C, therefore, lyophilization was implemented and validated to enable single use aliquots for improved consistency (Supplementary Fig. 2c).

**Leveraging the Strep-tag II for detection.** A widely used approach in immunology to detect antigen-specific T-cells is multimerization with Strepavidin[17]. Here, we aimed to tetramerize Siglec-Fc chimeras to generate octameric presentation. Streptavidin has an intermediate affinity for Strep-tag II ($K_d = 72\,\mu M$)[18], but the affinity of Strep-Tactin for Strep-tag II is 100-fold stronger[19]. Recombinant expression of Strep-Tactin[20] enabled incorporation of a formylglycine generating enzyme consensus site and subsequent site-specific fluorophore labeling (Supplementary Fig. 3a, b). AF647-labeled Strep-Tactin was produced and validated for detection of Siglec-1-Fc binding to cells (Supplementary Fig. 3c).

To test pre-complexing with Strep-Tactin, we used Siglec-1-Fc. Keeping the amount of Strep-Tactin constant (0.55 μg/mL; 40 nM of the monomer), the amount of Siglec-1-Fc was titrated from 0.6–60 μg/mL (4–400 nM) to cover a 1:10–10:1 molar ratio of Siglec-1-Fc dimer:Strep-Tactin monomer. Pre-complexing led to several orders of magnitude more binding to U937 cells compared to a two-step assay (Fig. 1c). Binding decreased above a 5:1 molar excess of Siglec-1-Fc, which may reflect excess Siglec-1-Fc outcompeting the pre-complexed material. At the optimal 5:1 ratio, no significant binding was observed for the R116A mutant of Siglec-1-Fc (Fig. 1d). Pre-complexing of the WT Siglec-Fc with Streptavidin led to very little binding (Supplementary Fig. 4a). Titrating the total complex at the optimal ratio, we found that sensitive binding to cells is achieved with 5–10 μg of Siglec-1 (Supplementary Fig. 4b). Comparing pre-complexing with Strep-Tactin-AF647 head-to-head with αhIgG1-AF647, Strep-Tactin pre-complexing showed superior sensitivity (Fig. 1e).

To assess the size of the pre-complexed Siglec, SEC was used with a high molecular weight separation range. Using Siglec-7-Fc, a complex of Siglec-7-Fc and Strep-Tactin eluted over a molecular weight range of 500 kDa–1.2 MDa (Supplementary Fig. 5a), which is in the range of four molecules of Siglec-7-Fc and

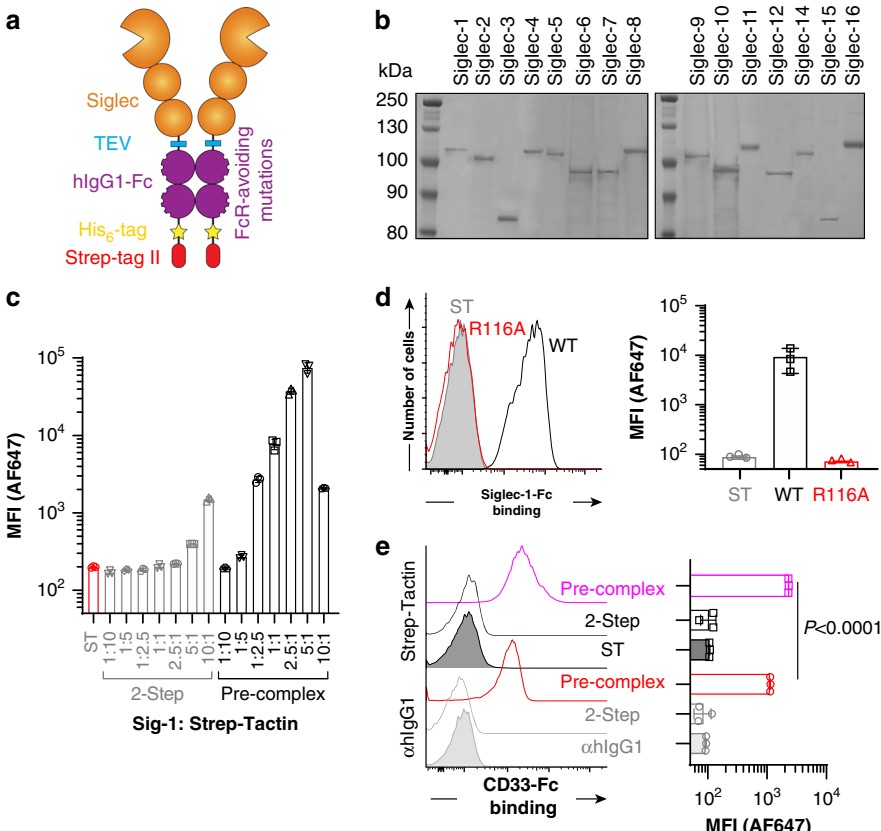

**Fig. 1 Production and validation of new Siglec-Fc proteins. a** Depiction of molecular features built into the new Siglec-Fc construct. **b** Coomassie-stained SDS-PAGE gel of all Siglecs purified through a three-step Nickel, ST, and SEC column purification strategy. Results are representative of two independent sets of purifications. **c** Different amounts of purified Siglec-1-Fc was incubated with U937 cells in either a two-step or a pre-complexed approach with a constant amount of Strep-Tactin-AF647. **d** Flow cytometry results for probing Siglec-1-Fc WT or Siglec-1-Fc R116A with U937 cells using the optimized pre-complexed conditions. **e** Flow cytometry results of probing CD33-Fc WT with U937 cells in a two-step assay or a pre-complexing assay with αhIgG1-AF647 or Strep-Tactin-AF647. Error bars represent ± standard deviation of three replicates. Statistical significance calculated based on a two-tailed unpaired Student's *t*-test.

Strep-Tactin (560 KDa + glycosylation). A smaller peak also appeared in the void volume, which may represent a small amount of aggregated protein. By SDS-PAGE, the isolated complex (elution volume 60–80 mL) clearly separated from excess Siglec-7-Fc (Supplementary Fig. 5b). The isolated complex showed strong binding to K562 cells and could be lyophilized and reconstituted without loss in cellular binding (Supplementary Fig. 5c, d).

**Profiling Siglec ligands on cells**. We next examined binding of pre-complexed Siglec-Fc chimeras with Strep-Tactin to a variety of cell lines that are representative of different origins. A549 cells were broadly recognized by many Siglecs (Fig. 2a). Each cell type displayed a different pattern of binding with all the Siglecs; HEK293T cells showed binding to Siglecs-1, -2, -5/14, -6, -7, and -8, K562 cells were bound by Siglecs-1, -2, -3, -7, and -9, U937 cells were bound by Siglecs-1, -2, -3, -5/14, -7, -9, and -10, Jurkat cells were recognized by Siglecs-1, -2, -7, and -9, A549 cells were recognized by Siglecs-1, -2, -3, -5/14, -7, -8, and -11/16, and HuH7 cells were recognized by Siglecs-1, -2, -3, -5/14, -7, and -8 (Fig. 2b and Supplementary Fig. 6). In all cases, the arginine mutants showed negligible binding. Sialic acid-dependent binding was confirmed through additional means; for example, Siglec-1, -2, and -7 binding to K562 cells was abolished by pre-treatment with neuraminidase (Supplementary Fig. 7a), while Siglecs-1, -2, and -7 failed to bind sialic acid-deficient CMAS$^{-/-}$ HEK293 cells (Supplementary Fig. 7b). As Siglec-12 did not show binding to any of the cell lines, it was not used in subsequent experiments.

**Profiling siglec ligands on primary human immune cells**. FcγR are abundant on immune cells and binding of Siglec-Fc proteins to cells can take place through FcγR instead of sialic acid. Binding of antibodies to FcγR can be mitigated with a Fc blocking agent, but αhIgG1 secondary antibody recognizes these blocking agents that are likely antibodies (Supplementary Fig. 8), consistent with previous observations[21]. To avoid FcγR interactions, a mutated version of hIgG1 was used in our constructs. Using the V1 and V2 of CD22-Fc, human splenocytes were probed to examine binding to neutrophils (Fig. 3a, b), which express low levels of α2-6 linked sialic acid and high levels of FcγRs[22], and B-cells (Fig. 3c), which express high levels of α2-6 linked sialic acid and lower levels of FcγR[23]. With V1, significant binding is observed to neutrophils with a reduction in the R120A mutant to levels that are still significantly above background (Fig. 3a). When Fc blocking agent was used, binding of both the WT and R120A mutant decreased, suggesting that binding to FcγRs contributes significantly to binding. The V2 of CD22-Fc showed overall lower binding, the reduced R120A mutant binding was much closer to background and was unaffected by Fc blocking agent (Fig. 3b). In contrast, V1 and V2 of CD22-Fc bind equally well to B-cells and was completely abrogated in the R120A mutant (Fig. 3c). These results confirm that V2 avoids binding to FcγRs.

Being confident that our Siglec-Fc chimeras pre-complexed with Strep-Tactin detect their sialic acid ligands on immune cells in a sensitive and selective manner, Siglec ligands were examined on human immune cells from the peripheral blood of three healthy individuals. Binding of each Siglec was assessed on

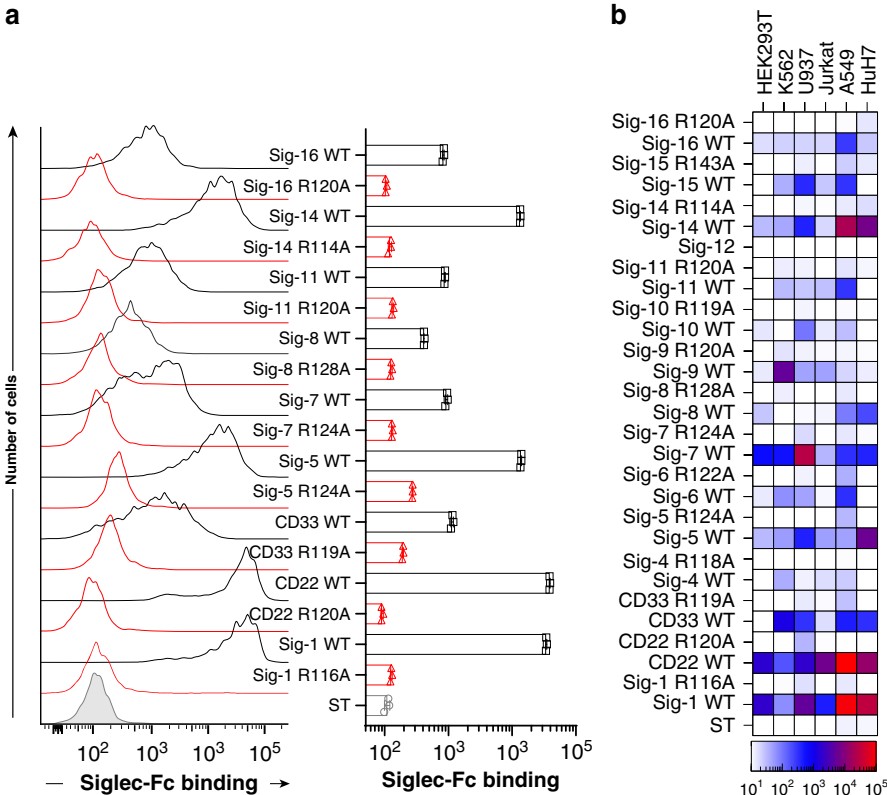

**Fig. 2 Using Siglec-Fc proteins to profile Siglec ligands on cell lines. a** Flow cytometry data of Siglec-Fc constructs pre-complexed with Strep-Tactin-AF647 binding to A549 cells. Error bars represent ± standard deviation of three replicates. **b** Heat map of binding of pre-complexed Siglecs to HEK293T, K562, U937, Jurkat, A549, and HuH7 cells.

B-cells, T-cells, NK cells, monocytes, neutrophils, basophils, plasmacytoid dendritic cells (pDCs), and conventional dendritic cells (cDCs) (Fig. 3d and Supplementary Figs. 9, 10). The highest binding for each Siglec was: Siglec-1 to monocytes, CD22 to B-cells, CD33 to neutrophils, Siglec-4 to pDCs, Siglec-6 to B-cells, Siglec-7 to basophils, Siglec-8 to pDCs, Siglec-9 to basophils, Siglec-10 to pDCs, Silgec-14 to B-cells, Siglec-15 to monocytes, and Siglec-16 to B-cells. To emphasize the sensitivity of detecting Siglec ligands on immune cells, a two-step and pre-complexed assay with Strep-Tactin was compared. As highlighted for binding of Siglec-1 to monocytes (Fig. 3e) there was a 3-fold increase in binding in the pre-complexed conditions, as well as a 5-fold increase for Siglec-7 to eosinophils (Fig. 3f), and a 4-fold increase for Siglec-9 to basophils (Fig. 3g). Using optimized pre-complexing conditions, we also examined Siglec ligands on human spleen sections by immunofluorescence (IF) microscopy. On the tissues, we observed staining of Siglec-7-Fc on all cell types (Fig. 3h), which was abrogated upon pre-treatment of the tissue by neuraminidase (Supplementary Fig. 11). The ability of Siglec-7-Fc to bind many cell types is consistent with broad staining of Siglec-7-Fc on splenocytes from the same spleen by flow cytometry (Supplementary Fig. 12).

**Probing Siglec ligands on cancerous tissues.** To identify Siglec ligands that are upregulated on cancer tissues, we initially probed seven breast cancer cell lines by flow cytometry (Fig. 4a and Supplementary Fig. 13). Of the Siglecs that showed binding, Siglec-7-Fc was one of the highest. The trend in binding of Siglec-7-Fc to cells by flow cytometry was also observed by IF microscopy (Fig. 4b). To determine how this translates to breast cancer tissues, we performed immunohistochemistry with Siglec-7-Fc pre-complexed with Strep-Tactin-HRP on six different breast

cancer tissue samples. Strong staining of Siglec-7-Fc to a number of breast cancer tissues was observed (Fig. 4c), but not with the arginine mutant (Fig. 4d). Consistent with our analysis of Siglec ligands on cancer cell lines, Siglec-7-Fc staining was expressed at high levels on several of the breast cancer tissues, but not all samples were high. Similar results were obtained for the same breast cancer tissues stained with Siglec-9-Fc (Supplementary Fig. 14).

**Preparing Siglec fragments with homogenous glycosylation.** Quantifying Siglec–ligand interactions in a way that provides equilibrium dissociation constants is challenging due to the low affinity of these interactions. Electrospray ionization mass spectrometry (ESI-MS) can measure protein–ligand complexes through detecting complexes in the gas phase, which report on equilibrium distributions in solution[24]. ESI-MS has been used to quantify weak affinities of protein-glycan interactions, such as those of the galectins[24]. To directly detect Siglec-glycan interactions by ESI-MS, a key obstacle was their large size and heterogeneity, owing to many glycoforms. To decrease their size, we generated monomeric Siglec fragments using TEV protease to remove the Fc. As demonstrated for Siglec-1-Fc, digestion with His$_6$-tagged TEV, followed by removal of both the TEV and Fc domain by Nickel column, yielded a highly purified Siglec-1 fragment (Fig. 5a). This approach was also effective in generating fragments of CD22 (Supplementary Fig. 15a) and CD33 (Supplementary Fig. 15b). To solve the issue of heterogeneity, we expressed the Siglec-Fc constructs in CHO Lec1 cells that produce high mannose N-glycans, which can be trimmed back to the first N-acetylglucosamine residue by Endo-H[25]. Following expression in Lec1 cells, removal of the Fc, and digestion with Endo-H, the mass spectra of Siglec-1-Fc went from exhibiting broad,

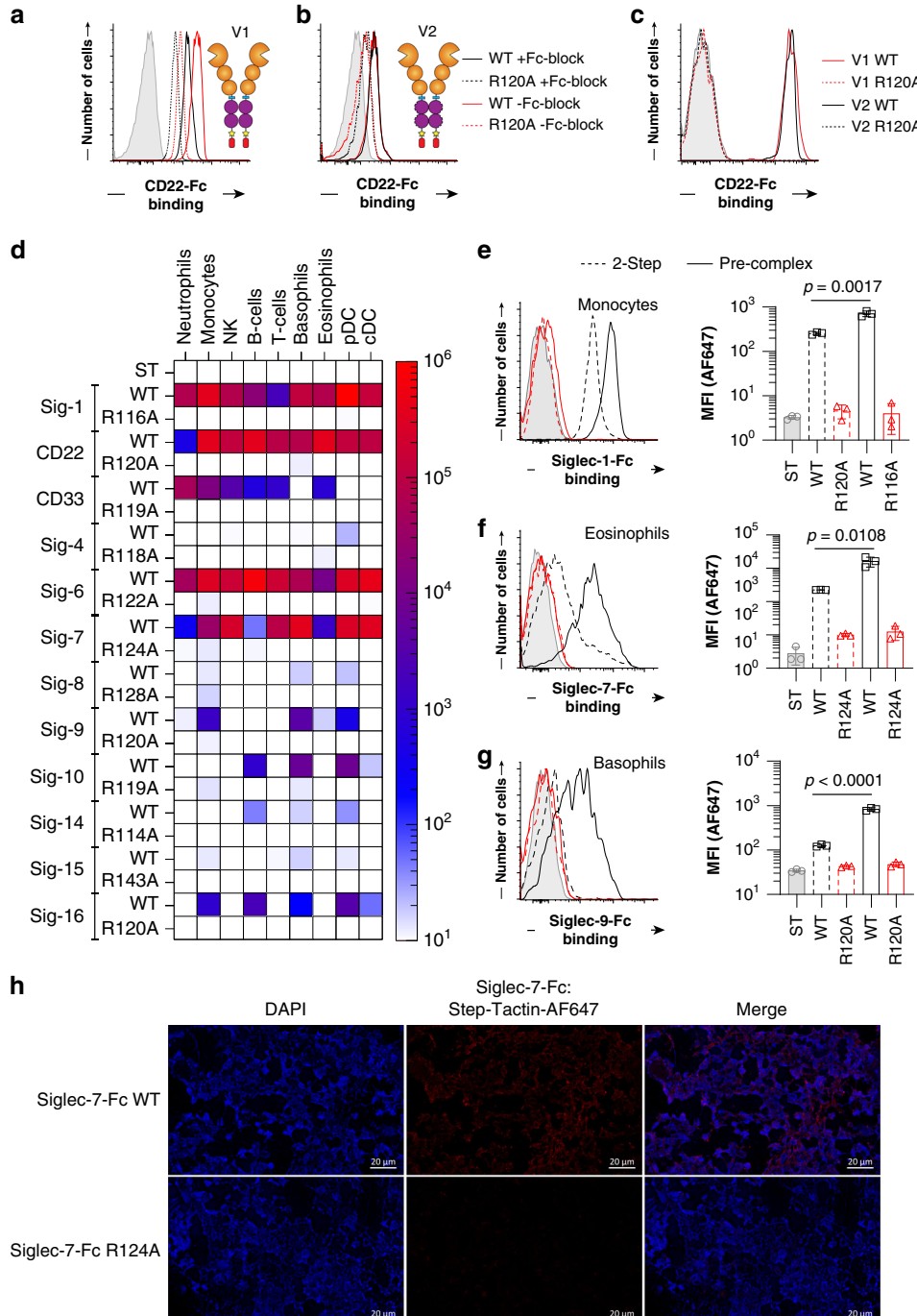

**Fig. 3 Profiling Siglec ligands on primary human immune cells. a** Binding of CD22-Fc V1 WT (red) and R120A mutant (black) to neutrophils that were pretreated with Fc receptor block (dashed) or without (solid) followed by staining with Strep-Tactin-AF647. Control staining (gray) is Strep-Tactin-AF647 alone. **b** Binding of CD22-Fc V2 WT (red) and R120A mutant (black) to neutrophils that were pretreated with Fc receptor block (dashed) or without (solid). **c** CD22-Fc V1 (red) or V2 (black) binding to B-cells. **d** Flow cytometry data of Siglec-Fc constructs that bound after pre-complexing with Strep-Tactin-AF647 to B-cells (CD19[+]), T-cells (CD3[+]), NK cells (CD56[+]), monocytes (CD14[+]), neutrophils (CD15[+]), eosinophils (Siglec-8[+]), basophils (CCR3[+]), pDCs (CD123[high]/HLA-DR[+]), and cDCs (CD123[dim]/HLA-DR[+]). **e–g** Binding to the indicated WT (black) and arginine mutant (red) Siglec-Fc to **e** monocytes, **f** eosinophils, and **g** basophils in either a 2-step assay (dashed) or pre-complexed assay (solid). **h** Immunofluorescence staining of human spleen with Siglec-7-Fc WT or Siglec-7-Fc R124A. Immunofluorescent staining was repeated three times with similar results. Error bars represent ± standard deviation of three replicates. Statistical significance calculated based on a two-tailed unpaired Student's *T*-test.

unresolved features at each charge state, to a sharp and resolved signal corresponding to the Siglec-1 fragment (Fig. 5b). For Siglec-1, the mass of the trimmed fragment (found:35,729.0 ± 0.2 Da) from Lec1 cells corresponds to the expected molecular weight of the polypeptide plus the addition of two HexNAc

residues (expected: 35,729 Da), consistent with two *N*-glycan sequons (Asn159,265). For the CD22 fragment, four peaks were observed (Fig. 5c and Supplementary Fig. 16a), corresponding to the expected masses of the trimmed CD22 fragment with three or four HexNAc residues (predicted *N*-glycan sites at

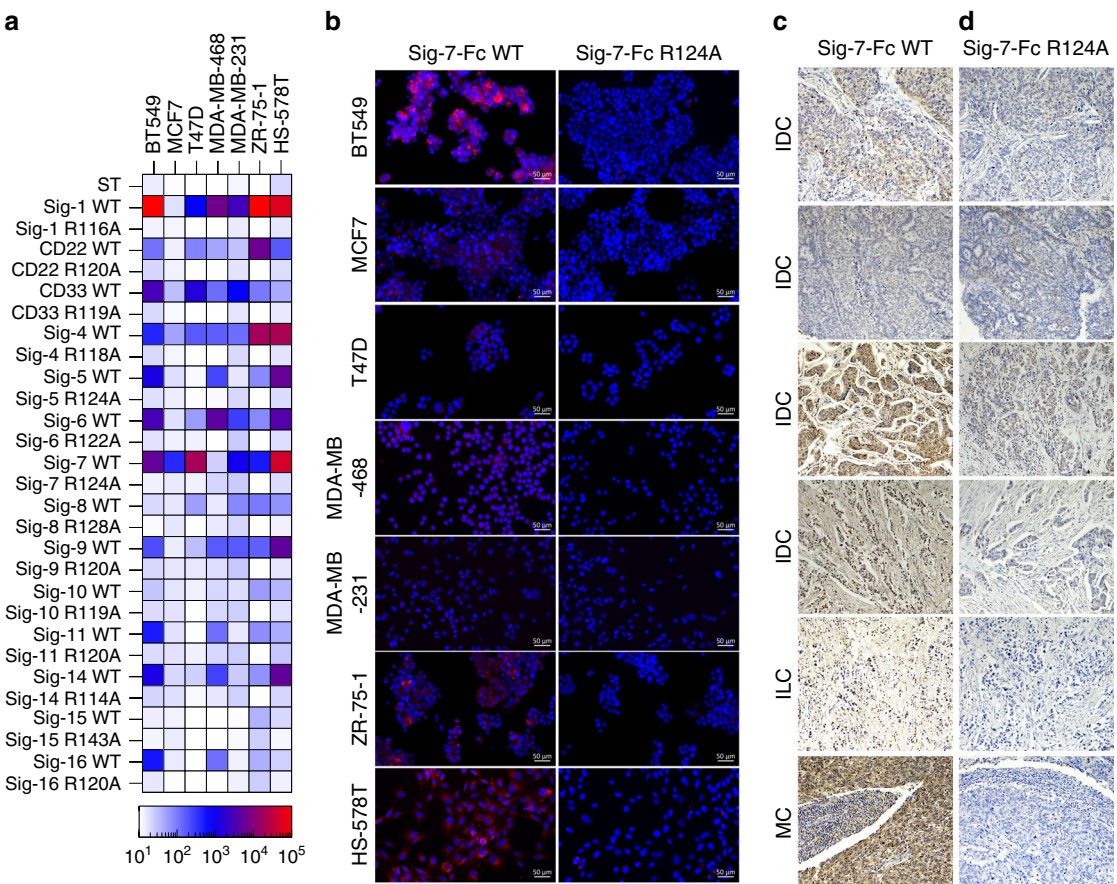

**Fig. 4 Siglec ligands on cancer cells and tissues. a** Heat map of binding of pre-complexed Siglecs to breast cancer cell lines determined by flow cytometry. **b** Immunofluorescence staining of cancer cell lines BT549, MCF-7, T47D, MDA-MB-468, MDA-MB-231, ZR-75-1, and HS-578T with Siglec-7-Fc WT and R124A. Immunofluorescence results on breast cancer cell lines are representative of three independent replicates. **c, d** Breast cancer patient tissue cores with a pathology diagnosis of IDC (invasive ductal carcinoma), ILC (invasive lobular carcinoma) and MC (medullary carcinoma) were immunostained with: Siglec-7-Fc WT (**c**), Siglec-7-Fc R124A (**d**) and imaged at ×20 magnification. Positive signal is represented by brown coloration from chromogenic staining. Darker staining as represented by increased brown coloration indicates Siglec-7-Fc binding (**c**) as compared to control (**d**), while blue staining represents cell nuclei (hemotaxylin stain). IHC results are representative of two independent replicates. Scale bars are indicated as 20 μm with a tissue width of 670 μm.

Asn67,101,112,135,164), with and without one core fucose. For the CD33 fragment, three peaks were observed (Fig. 5d and Supplementary Fig. 16b), corresponding to the expected mass of the CD33 fragment with five HexNAc residues (*N*-glycan sequons at Asn100,113,160,209,230). Unexpectedly, 50% of CD33 contained an *O*-glycan, which was either a sialyl-T or disialyl-T antigen. While some small degree of heterogeneity remained for CD22 and CD33, peaks were sufficiently resolved to perform binding studies.

**Quantifying Siglec–glycan interactions by mass spectrometry.** ESI-MS was used to detect binding of a number of sialosides to several Siglecs. We began by measuring the affinity of Siglec-1, CD22, and CD33 with four sialosides: Neu5Acα2-6Lac (1), Neu5Acα2-3Lac (2), Neu5Acα2-6LacNAc (3), and Neu5Acα2-3LacNAc (4). Shown in Fig. 5e is an example of binding for Siglec-1 to Neu5Acα2-3Lac (2), where the complex is clearly resolved. Titrating the concentration of ligand enabled the $K_D$ values to be determined (Fig. 5f and Supplementary Fig. 17–19). Siglec-1 recognized Neu5Acα2-3Lac (2) the strongest ($K_D$ = 0.5 mM), with a small preference for the underlying lactose over LacNAc and α2-3 over α2-6. CD22 recognized Neu5Acα2-6LacNAc (3) the strongest ($K_D$ = 0.06 mM) with a small preference for an underlying LacNAc and absolute requirement for

an α2-6 linkage. CD33 recognized all four sialosides with comparably weak binding ($K_D$ = 2-3 mM). We also examined binding of CD33 to a chemically-modified sialoside (5) previously designed to be selective for CD33[26] (Supplementary Fig. 20a). Titrating this ligand and determining the percentage of CD33-ligand complex enabled a $K_D$ value of 87 ± 3 μM to be calculated (Supplementary Fig. 20b, c). When CD33 R119A was examined for binding towards compounds 1–5, no protein–ligand complex was observed relative to non-specific binding accounted for by the reference protein (Supplementary Fig. 21).

**Probing the cellular ligands of CD33.** The mass spectrometry results suggest that CD33 recognizes both α2-3 to α2-6 sialosides with similar affinities, while plate-based results suggest a preference for α2-6 sialosides[27]. To clarify this point, we examined *trans* and *cis* CD33 ligands on monocytic U937 cells and primary monocytes using two strategies. To detect *trans* ligands, we used pre-complexed CD33-Fc and observed significantly decreased staining of U937 cells following loss of α2-3-linked sialic acid with neuraminidase-S (Neu-S), and a complete loss when cells were treated with a neuraminidase that destroys both α2-3 and α2-6 sialosides (neuraminidase-A; Neu-A) (Fig. 6a). CD33-Fc binding was also impaired to ST6Gal1$^{-/-}$ U937 cells relative to WT cells, and completely abrogated when ST6Gal1$^{-/-}$ cells were treated

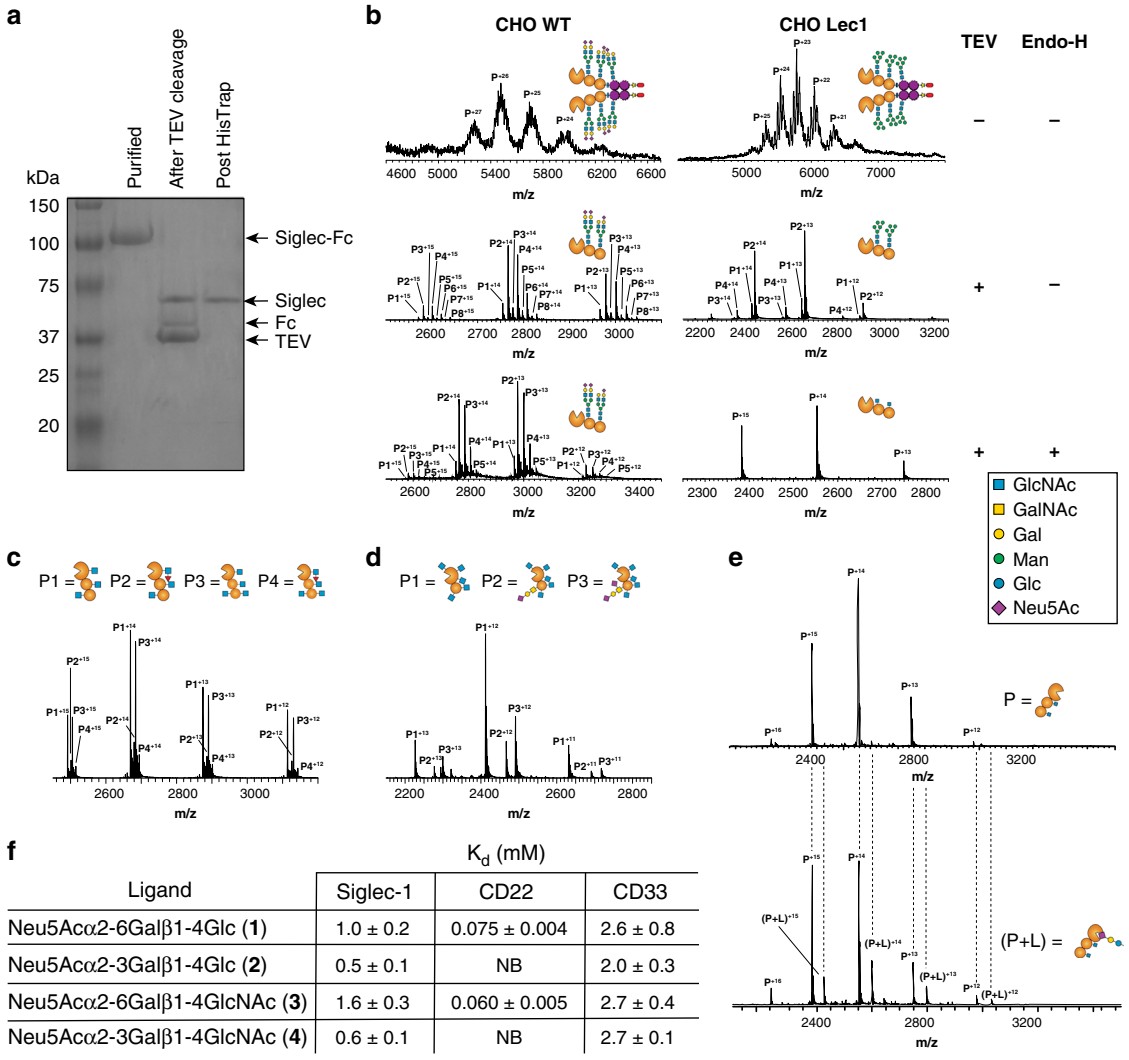

**Fig. 5 A mass spectrometry assay to detect and quantify Siglec-glycan interactions. a** SDS-PAGE of Siglec-1-Fc following digestion with TEV and Endo-H. **b** Mass spectrum for Siglec-1 produced from WT and Lec1 CHO cells as the full Fc-chimera, following digestion with TEV, and after treatment with Endo-H. **c**, **d** Mass spectrum of CD22 (**c**) and CD33 (**d**) from Lec1 CHO cells treated with TEV and Endo-H. **e** ESI mass spectra of 3.6 μM Siglec-1 in its unliganded form (upper) and complexed with 80 μM Neu5Acα2-3Lac (2) (lower). Spectra are representative of at least two independent replicates. Dashed lines are a guide to line up the same speak in each spectrum. **f** Summary of affinities ($K_D$) of the four trisaccharides for the fragments of Siglec-1, CD22, and CD33.

with Neu-S (Fig. 6b). In contrast, Siglec-1-Fc binding to U937 cells was completely dependent on α2-3 sialosides (Supplementary Fig. 22a, b) while CD22-Fc binding was completely dependent on α2-6 sialosides (Supplementary Fig. 22c, d). Similar results were observed for binding of CD33-Fc to human blood CD14[+] monocytes treated with Neu-S or Neu-A (Fig. 6c). We next examined *cis* ligands of CD33 through an unmasking assay[28] using fluorescent liposomes bearing CD33 ligand (5) conjugated to a lipid (6) (Fig. 6d and Supplementary Fig. 23). Neu-S unmasked CD33 on both U937 cells and peripheral blood monocytes (Fig. 6e, f), as evidenced by significantly increased binding to the CD33L liposomes. Further unmasking was observed with Neu-A (Fig. 6f). These results strongly suggest that both α2-3 and α2-6 sialosides serve as cellular ligands of CD33.

## Discussion

Interactions between Siglecs and their sialic acid-containing glycan ligands are proposed to be a form of 'self' recognition used by immune cells. Indeed, sialic acid is not expressed by many pathogens and those that do may exploit Siglecs in a similar

manner as cancer cells for the purposes of dampening immune cell responses. These roles have led Siglecs to be considered immune checkpoints[3,11], and efforts to block these checkpoints are underway[29,30]. The consequence of blocking Siglec–ligand interactions in vivo is unknown because of an incomplete description of the nature and location of their glycan ligands. Motivated to better understanding Siglec ligands as self-associated molecular patterns, we have developed a new set of soluble Siglecs to probe Siglec-glycan interactions.

Most Siglec–ligand interactions described to date are relatively weak affinity, with $K_D$ values in the range of 0.05–5 mM[15]. Studying these interactions on cells and tissues is achieved through increasing multivalency and avidity. Fc chimeras of glycan-binding proteins, including Siglecs[31] and C-type lectins[32], have been widely used. The dimeric nature of Fc chimeras is sufficient, in some cases, to achieve binding, but further increasing multivalency is expected to be advantageous. Several efforts have been made to increase multivalency of Siglecs. For example, biotinylation of monomeric Siglec-7 to create tetramers with Streptavidin[33], pre-complexing Siglec-Fcs with αhIgG1[12], and pentameric *comp* constructs of Siglecs to detect Siglec-8 and

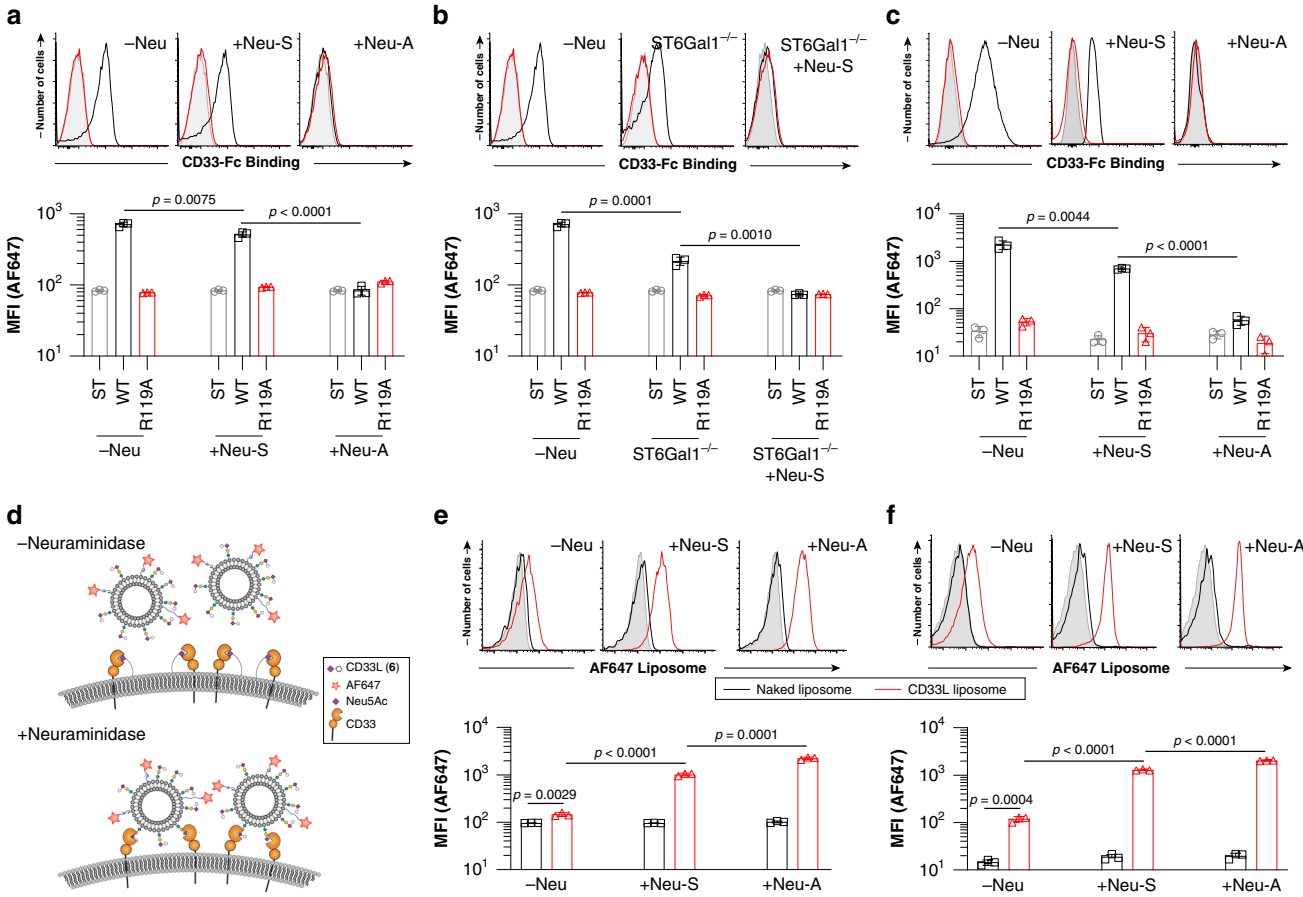

**Fig. 6 Cellular ligands of CD33 on human monocytes are both α2-3 and α2-6 sialosides. a** Staining of U937 cells treated with an α2-3-specific sialiadase (Neu-S) or broadly acting sialiadase (Neu-A) with CD33-Fc pre-complexed with Strep-Tactin-AF647. **b** Staining of WT and ST6Gal1$^{-/-}$ U937 cells with CD33-Fc pre-complexed with Strep-Tactin-AF647. **c** CD33-Fc binding to human peripheral blood monocytes treated with Neu-S or Neu-A. **d** Depiction of the unmasking assay performed using CD33 high-affinity ligand (CD33L, compound 6) displaying liposomes on WT cells or cells treated with neuraminidase. **e, f** Binding of CD33 ligand-targeted liposomes (red) or naked liposomes (black) to U937 cells (**e**) or to human peripheral blood monocytes (**f**). Error bars represent ± standard deviation of three replicates. Statistical significance calculated based on a two-tailed unpaired Student's T-test.

-9 ligands[13]. Taking this a step further, we have demonstrated that tetramerization of dimeric Siglec-Fc chimeras provides greatly enhanced sensitivity for detecting sialic acid ligands on cells and tissues. In addition to enhanced sensitivity for binding cellular ligands, our constructs have features that make them selective for their glycan ligands. Avoiding a αhIgG1 secondary through use of Strep-Tactin and a mutated Fc to avoid interactions with FcγRs. Moreover, each Siglecs-Fc construct has a corresponding mutant lacking the essential arginine that serves as an excellent control.

Siglecs bind to only a subset of sialosides in the glycome owing to their specificity for the type of sialoside linkage or underlying glycan. Consistent with this, all Siglecs—except Siglec-5/14 and 11/16, which share an identical V-set domain with their pair, as well as Siglec-12 that lacks sialic acid binding—showed a unique binding pattern to cells. For example, we observed strong binding of Siglec-7-Fc to NK cells, which is consistent with previous findings demonstrating that Siglec-7 is highly masked on NK cells[34]. Siglecs-11/16 are reported to bind poly sialic acid[35], and the binding of Siglec-11/16 was observed on A549 and several breast cancer cell lines, which is intriguing since the upregulation of polysialic acid on cancer cells is an area of emerging interest[36]. Interestingly, Siglec-8 did not bind to primary immune cells but did show significant binding to several cancer cell lines. Siglec-8 prefers a sulfate group on the underlying glycan[37], and sulfated glycans can be upregulated in cancer[38].

Interactions between Siglecs and glycan ligands is commonly studied with plate-, bead-, and microarray-based assays[39,40]. In plate- and bead-based approaches, competitive assays yield $IC_{50}$ values that do not necessarily reflect $K_D$ values. A case in point is the previously designed high affinity and selective CD33 ligand[26], where a flow-cytometry competitive bead binding assay provided an $IC_{50}$ value of 11 μM, but later a $K_D$ of 118 μM was measured by surface plasmon resonance (SPR)[41]. This latter value is in line with the $K_D$ value of 87 μM measured by the ESI-MS binding assay. One key advantage of the mass spectrometry approach developed here to study Siglec–ligand interactions, compared to other approaches such as ITC and SPR, is that it allows very weak interactions to be directly quantified with minimal amounts of the Siglec protein. Using this assay, several notable observations were made about the ligands of Siglecs. For Siglec-1, we observed specificity towards α2-3 sialosides over α2-6 sialosides, for CD22 an absolute requirement for an α2-6 linkage, and CD33 appeared to bind both α2-6 and α2-3 sialosides equally. Specificities and affinities of Siglec-1 and CD22 are consistent with previous measurements[42]. For CD33, there have been conflicting reports on its specificity. An older study using re-sialylation of erythrocytes suggested a preference for α2-3 sialosides[43] and a newer glycan microarray suggested a lack of preference[44]. On the other hand, two influential plate-based array studies suggested a preference for α2-6 sialosides[27,42]. Our ESI-MS assay revealed that CD33 bound α2-3 sialosides or α2-6 sialosides with equal affinity,

therefore, we leveraged our new CD33-Fc constructs to probe the cellular ligands of CD33 ligands, which we find to be both α2-3 and α2-6 sialosides. These findings may have important implications to human disease, since an alternative isoform of CD33 lacking its glycan binding, which is enhanced by a rare allele in humans, correlates with Alzheimer's disease (AD) susceptibility[45].

In summary, our new Siglec-Fc constructs have the ability to detect Siglec–ligand interactions in a sensitive, selective, and quantitative manner. Siglec ligands were profiled on healthy human immune cells and tissues, as well as cancerous cells and tissues. Used in combination with cell-based glycan arrays[46,47], we anticipate that these Siglec-Fc proteins will be of great use in elucidating finer details about the glycan ligand specificity of Siglecs as self-associated molecular patterns.

## Methods

**Human spleen and peripheral blood studies.** All human studies were performed with informed consent and approved by the Health Research Ethics Board—Biomedical Panel at the University of Alberta.

**Breast cancer tissues.** Breast cancer tissue arrays were obtained from US Biomax. Study approval was obtained by the institutional review board of UBC (IRB#H17-01442). Study is compliant with all relevant ethical regulations on the use of human tissues.

**Statistics and reproducibility.** A student's $t$ test was used to assess statistical significance. All assays were conducted with replicates of $n = 3$ and human peripheral blood experiments were conducted with $n = 3$ of 3 patients.

**General methods.** Primers were obtained from Integrated DNA Technologies (IDT). PCR, site-directed mutagenesis reagents, and restriction enzymes were obtained from New England Biolabs. PCR purification and Mini Prep kits were from GeneAid. Sanger sequence was performed through the University of Alberta Molecular Biology Service Unit. Cell culture reagents were obtained from Gibco (Thermo Fisher).

**Cloning of the hIgG1 Fc vector.** A fragment encoding human IgG1 Fc without (version 1) or with (version 2) 7 mutations (L234A, L235A, G237A, H268A, P238S, A330S, and P331S) and along containing an N-terminal TEV cleavage site sequence (ENLYFQG) and C-terminal His_6 and Strep-Tag II (WSHPQFEK) was synthesized by GeneArt (Thermo Fisher). The two versions were PCR-amplified with a 5′ AgeI restriction site and 3′ AvaI restriction site, and doubly digested with AgeI and AvaI. The digested PCR products were ligated into the pcDNA5/FRT/V5-His-TOPO® vector cut with AgeI. Sanger sequencing was used to select a version where the insert was ligated in the correct orientation to allow for in frame cloning of Siglecs using the NheI and AgeI sites in the next step.

**Cloning of siglec-Fc constructs.** The genes encoding human Siglecs 1-16 (there is no Siglec-13 in humans) were synthesized from GeneArt and provided in pMX cloning vectors with a 5′ NheI site and 3′ AgeI site. For a several Siglecs that had internal cut sites, silent mutations were incorporated to ablate the restriction site. PCR was performed to obtain the two or three most N-terminal extracellular domains of each Siglec with a 5′ NheI and 3′ AgeI cut site using the primers used are listed in Supplementary Table 2. The PCR products were ligated with the hIgG1 Fc vector (see above). Sanger sequencing was performed to validate each construct.

**Site-directed mutagenesis.** Using the mutagenesis primers shown in Supplementary Table 3, site-directed mutagenesis was performed on all the human Siglecs to remove the essential arginine that is required for binding sialic acid. Following the recommended protocol by New England Biolabs, the mutagenesis product was digested with DpnI restriction enzyme, heat inactivated, and transformed into ultracompetent *E. coli*. Sanger sequencing was used to verify successful mutagenesis. The mutant constructs were subsequently subcloned in the Fc-containing vector as described above.

**Generation of CMAS$^{-/-}$ 293 cells and ST6Gal1$^{-/-}$ U937 cells.** Custom crRNA (IDT) was designed to target human CMAS (GAACACCCCCGATCTTCTCC) and ST6Gal1 (CAGATGGGTCCCATACAATT). Cells were seeded at 500,000 cells per well 1 day prior to transfection in a 6-well tissue culture plate. For one well of a 6-well plate, 20 pmol of Cas9 nuclease (IDT), 20 pmol of ATTO-550 labeled crRNA:tracrRNA (IDT) duplex, 8 μL Cas9 Plus reagent (IDT), and 16 μL CRISPRMAX reagent (Thermo Fisher) in 600 μL Opti-MEM medium (Gibco). One day after transfection, cells were removed from the plate by trypsin digestion,

washed, resuspended in 300 μl of cell sorting medium (HBSS, 10% FBS, 1 mM EDTA), and stored on ice until sorting. Cells were sorted within the University of Alberta Flow Cytometry Core. The top 5% bright dyes stained with ATTO-550 were sorted into three 96-well plates containing regular culture medium at one cell per well. Cells were grown for ~2 weeks until a time when colonies were screened by flow cytometry using fluorescein-conjugated Sambucus Nigra Lectin (SNA, 1:750, Vector Laboratories).

**Stable transfection of siglec-Fc constructs.** Flp-In CHO WT and Flp-In Lec1 CHO cells were cultured in complete medium (DMEM/F12 supplemented with 10% Fetal Bovine Serum (FBS; Gibco) and 100 U/ml Penicillin, and 100 μg/ml Streptomycin (Gibco)) within a 5% 37 °C CO$_2$ incubator. To transfect, cells were seeded at 375,000 cells/well in a 12-well plate. After 24 h, 1.8 μg of pOG44 and 0.2 μg of DNA to be transfected, and 2.35 μL of Lipofectamine PLUS reagent (Thermo Fisher) was added to 0.25 mL of Opti-MEM (Thermo Fisher) in a 1.5 mL microcentrifuge tube. After 15 min, 7.85 μL of Lipofectamine LTX reagent (Thermo Fisher) was added to the tube. After 30 min, the mixture was added to cells that were washed with Opti-MEM and placed in a 5% 37 °C CO$_2$ incubator for 24 h at which time 1 mL of complete medium is added to the well. After 48 from initial transfection, cells are removed with 5 mM EDTA/PBS and re-plated onto a T-25 flask in complete medium with 0.5 mg/mL of Hygromycin B (Thermo Fisher). Further selection of transfected cells was completed by changing the complete media with 0.5 mg/mL of Hygromycin B being changed every 2 days for 2 weeks. ST6Gal1-expressing CHO cells were made in the same way by transfecting human ST6Gal1 in the pcDNA5/FRT vector into Flp-In CHO cells.

**Siglec-Fc protein expression and quantitation by ELISA.** Cells were split between 1:5 to 1:10 into T-175 flasks with 50–60 mL of media supplemented with 10 mM HEPES. Once cells reached confluency, the supernatant was harvested 7 days later by filtering the supernatant through a 0.22 μm filter. The filtered supernatant was stored at 4 °C until ready for use or purification. To estimate the amount of Siglec-Fc in the supernatant, an ELISA was carried out by coating wells overnight with anti-human IgG (Thermo Fisher) in phosphate-buffered saline (PBS). The following day, plates were washed in PBS-T (PBS containing 0.1% Tween-20) and incubated with PBS containing 5% BSA for 45 min. After five washes with PBS-T, the plate was then incubated for 1 hour with serial dilution of the supernatants. Serial dilutions were made in supernatants expressed from CHO Flp-in cells that had not been transfected, which showed no signal by ELISA. Following washing, wells were incubated for 1 h with Strep-Tactin-HRP (IBA Lifesciences) at a concentration of 0.1 μg/mL in PBS-T containing 1% BSA. After five washes of PBS-T, wells were developed with TMB Microwell Peroxidase Substrate System (KPL) for 15 min and quenched with 1 M phosphoric acid and read at 450 nm. Purified protein (see below) was used to generate a standard curve to estimate the amount of protein in the supernatants.

**Siglec-Fc protein purification.** Supernatant was purified by Ni$^{2+}$-Affinity chromatography by equilibrating a 1 mL or 5 mL HisTrap Excel (GE) with 15 column volume (CV) of equilibrium buffer (20 mM sodium phosphate, 0.5 M NaCl, pH 7.4). Supernatant was then loaded onto the column, followed by 15 CV of washing buffer (20 mM sodium phosphate, 0.5 M NaCl, 30 mM imidazole, pH 7.4). To elute the protein, 15 CV of elution buffer (20 mM sodium phosphate, 0.5 M NaCl, 500 mM imidazole, pH 7.4) was added and 1 mL fractions were collected. Fractions that contain protein (identified by A280 values taken on the NanoDrop (Thermo Fisher)) were combined and diluted 10-fold with buffer W (100 mM Tris-HCl, 150 mM NaCl, 1 mM EDTA, pH 8). A 1mL Strep-Tactin column (IBA Life-Sciences) was equilibrated with 5 CV of buffer W and then the diluted protein sample was loaded onto the column. After, the column was washed with 5 CV of buffer W and then the protein was eluted with 10 CV of buffer E (100 mM Tris-HCl, 150 mM NaCl, 1 mM EDTA, pH 8, 5 mM Desthiobiotin) into 1 mL fractions. Fractions that contain protein were combined and concentrated down to 1 mL by Amicon Ultra Centrifugal Filters (Thermo Fisher) and then loaded onto a HiPrep 16/60 Sephacryl S-100HR equilibrated with degassed Column Buffer (250 mM NaCl, 10 mM Tris-HCl, pH 7.5) at 4 °C. Protein fractions were collected using a flow rate of 0.5 mL/min. Protein-containing fractions were pooled and concentrated to 0.12 mg/ml (Amicon Ultra-15 Centrifugal Filter Unit, 10 kDa molecular weight cut-off). Resulting samples were either stored at 4 °C or lyophilized in small aliquots and stored at −20 °C.

**Bacterial protein expression**
*Tobacco Etch Virus (TEV) protease.* The gene encoding TEV with a N-terminal His_6-tag was transformed into BL21 pLysS competent cells was grown in 30 mL of LB media with 100 mg/L ampicillin and 25 mg/L chloramphenicol overnight at 37 °C. The following morning, 10 mL of the starter culture was added to 1 L of LB media with the same ratio of antibiotic and allowed to grow at 37 °C until OD$_{600}$ = 1.5. After, 1 mM IPTG was added and the culture was shaken at 18 °C for 18 h. The culture was spun down at 14,000 rcf for 30 min at 4 °C, after which the pellet was resuspended in lysis buffer (20 mM Tris, 500 mM NaCl, 5 mM Imidazole, 5% Glycerol, 1 mM DTT, pH 8.0) and cells were lysed using a tissue homogenizer. The eluted solution was filtered using a 0.45 μm PES filter unit (Fisher Scientific) and

loaded onto a HisTrap HP (GE LifeSciences) that was equilibrated with 10 CV of lysis buffer. The protein was eluted off of the column using 10 CV of elution buffer (20 mM Tris, 500 mM NaCl, 500 mM Imidazole, 5% glycerol, 1 mM DTT, pH 8.0) and then buffer exchanged into storage buffer (20 mM Tris, 20% glycerol, 1 mM DTT, pH 8.0).

*Formylglycine generating enzyme (FGE)*. N-terminally His$_6$-tagged FGE (Addgene plasmid #75147[48]) was transformed into BL21 competent cells and grown overnight in 5 mL of LB media with 100 mg/L ampicillin (Thermo Fisher) at 37 °C. The next day, 1 L LB media with 100 mg/L ampicillin was inoculated with 2 mL of the starter culture and grown at 37 °C until O.D. = 0.6. The flask was set in ice for 10 min, after which 100 µM IPTG was added and the culture was allowed to continue growing at 37 °C for 20 h. The culture was then spun down at 5000 rcf for 20 min. The resulting pellets were resuspended in 50 mL of lysis buffer (DPBS, 150 mM NaCl, 10 mM imidazole, 1 mM TCEP, pH 7.4) with 500 µL of Halt Protease Inhibitor Cocktail (100x) (Thermo Fisher) and 5 µL of Pierce Nuclease (Thermo Fisher). The solution was shaken at 3 rcf for 2 h at 4 °C then the cells were lysed through a tissue homogenizer. The eluted solution from the homogenizer was spun down at 9000 rcf for 30 min. The supernatant was taken and loaded onto a 1 mL HisTrap HP (GE LifeSciences) that was equilibrated with 10 CV of lysis buffer. The column was washed with 10 CV of washing buffer (DPBS, 150 mM NaCl, 20 mM imidazole, 1 mM TCEP, pH 7.4), then the pure FGE protein was eluted with 10 CV of elution buffer (DPBS, 150 mM NaCl, 250 mM imidazole, 1 mM TCEP, pH 7.4). The protein was quantified using A280 values taken on the NanoDrop (Thermo Fisher), and then buffer exchanged into triethanolamine buffer (25 mM triethanolamine, 50 mM NaCl, pH 9). Unused protein was frozen into aliquots by the addition of 2 mM β-mercaptoethanol and 10% glycerol and stored in −80 °C.

*Strep-Tactin*. The gene encoding Step-Tactin with a tandem FGE consensus and His$_6$-tag on the C-terminal side of the signal sequence was synthesized by GeneArt into pRSET_A_A185 vector similar to another report for preparation of His6-tagged Streptavidin[49]. FGE-His$_6$-Strep-Tactin was transformed into BL21 pLysS competent cells was grown in 30 mL of LB media with 100 mg/L ampicillin and 25 mg/L chloramphenicol overnight at 37 °C. The following morning, 10 mL of the starter culture was added to 1 L of LB media with the same ratio of antibiotic and allowed to grow at 37 °C until OD$_{600}$ = 1.0. After, 1 mM IPTG was added and the culture was shaken at 18 °C for 24 h. The culture was then spun down at 5000 rcf for 20 min and filtered using a 0.45 µm PES Filter Unit (Fisher Scientific) and loaded onto a HisTrap HP (GE LifeSciences) that was equilibrated with 10 CV of equilibrium buffer (20 mM sodium phosphate, 0.5 M NaCl, pH 7.4). The column was washed with 10 CV of washing buffer (20 mM sodium phosphate, 0.5 M NaCl, 30 mM imidazole, pH 7.4), then the protein was eluted with 10 CV of elution buffer (20 mM sodium phosphate, 0.5 M NaCl, 500 mM imidazole, pH 7.4). The protein was quantified using A$_{280}$ values taken on the NanoDrop (Thermo Fisher), and then buffer exchanged into triethanolamine buffer (25 mM triethanolamine, 50 mM NaCl, pH 9).

*Neu-A and Neu-S*. The coding sequence of Neuraminidase-A and Neuraminidase-S (GenBank accession number: AY934539.2 for Neu-A and ABJ55283 for Neu-S) were commercially synthesized into pET100/D-TOPO vectors. (GeneArt Gene Synthesis, CA) The pET100/D-TOPO vectors containing each sialidase gene were chemically transformed into BL-21 competent cells (New England Biolabs, CA). The transformed cells were cultured in 50 mL of lysogeny broth (LB) containing 100 µg/mL of ampicillin at 37 °C for 18 h on the shaking incubator with 20 rcf. The cultured cells were scaled up to 1 L of LB-amp for 4 h. After OD$_{600}$ reached 0.6, 1 mM of IPTG was added and the culture was shaken at 25 °C for 18 h. The culture was centrifuged at 13,000 rcf for 30 min. The cell pellet was resuspended in 40 mL of cell lysis buffer (50 mM of Na$_x$PO$_4$, 400 mM NaCl, 0.1% w/v of triton-100, 1 mg/mL of lysozyme, pH 7.7) and incubated at 37 °C for an hour. Then, the cells were sonicated with half intensity of 50% pulses with three intervals of 10 s sonication and 50 s rest. The cell lysates were centrifuged at 18,000 rcf for 30 min. The supernatant was filtered through a 0.22 µm filter and loaded onto a 1 mL HisTrap HP (GE LifeSciences) that was equilibrated with 10 CV of lysis buffer. The column was then washed with washing buffer (50 mM sodium phosphate, 400 mM NaCl, pH 7.7), and then eluted with elution buffer (50 mM sodium phosphate, 400 mM NaCl, 500 mM of imidazole, pH 7.7). The elution fractions containing proteins were pooled together and dialyzed three times with a storage buffer (50 mM NaCl, 20 mM Tris-HCl, 1 mM EDTA, pH 7.5). After dialysis, each aliquot was stored at −80 °C until use.

**TEV cleavage of soluble Siglec-Fc proteins**. Purified Siglec-Fc was incubated with 10 times molar excess of TEV protease at 30 °C for 8 h. SDS-PAGE was used to verify that no intact Siglec-Fc remained. The reaction mixture was purified by passing it over a HisTrap Excel (GE LifeSciences) preequilibrated with 15 CV of equilibrium buffer (20 mM sodium phosphate, 0.5 M NaCl, pH 7.4) and washed with 5 CV of washing buffer (20 mM sodium phosphate, 0.5 M NaCl, 30 mM imidazole, pH 7.4). The flow-through was collected and buffer exchanged into 200 mM ammonium acetate for mass spectrometry.

**Site-specific modification of strep-tactin**. The Strep-Tactin was incubated with a 10% molar ratio of FGE enzyme in the presence of TEAM buffer with 5 µM CuSO4

and 2 mM BME final concentration. It was allowed to shake gently at 30 °C overnight. The FGE was purified away from the Strep-Tactin by passing the reaction mixture over a Sephadex G-25 column (GE LifeSciences). The resulting Strep-Tactin protein was added to 2 mM AF647-hydroxylamine (Thermo Fisher) in 200 mM anilininum acetate, pH 4.7 at a 1:1 v/v ratio and allowed to react for 1 h at rt. The reaction mixture was purified using a Sephadex G-25 column to separate unreacted AF647 away from the Strep-Tactin-AF647. The solution was concentrated down to 1.0 mg/mL and stored at 4 °C.

### Flow cytometry detection of siglec ligands on cells
*2-Step assay with Strep-Tactin, streptavidin, or anti-human IgG1*. Siglec supernatant (50 µL) or purified Siglec was incubated with cells in a 96-well U-bottom plate for 30 min at 4 °C. Following the incubation, 200 µL of flow buffer (HBSS, 0.1% BSA, 5 mM EDTA) was added and the plate was spun down at 300 rcf for 5 min. Strep-Tactin-AF647 (0.55 µg/mL, 50 µL, 0.57 pmol), streptavidin (Biolegend), or anti-human IgG1 (Biolegend) was added to the cells and incubated for 30 min at 4 °C. After, 200 µL of flow buffer (HBSS, 0.1% BSA, 5 mM EDTA) was added and the plate was spun down at 300 rcf for 5 min. The cells were resuspended in 200 µL of flow buffer and processed through the flow cytometer.

*Pre-complexing with Strep-Tactin, streptavidin or anti-human IgG1*. Strep-Tactin-AF647 (0.55 µg/mL, 0.57 pmol), streptavidin (Biolegend), or anti-human igG1 (Biolegend) was added to Siglec supernatant (50 µL) or purified Siglec and allowed to sit at 4 °C for 15 min. After, the pre-complexed solution was added to cells in a 96-well U-bottom plate for 30 min at 4 °C. After, 200 µL of flow buffer (HBSS, 0.1% BSA, 5 mM EDTA) was added and the plate was spun down at 300 rcf for 5 min. The cells were resuspended in 200 µL of flow buffer and processed through the flow cytometer.

**Analysis of siglec ligands on human immune cells**. For human splenocytes, the sample was homogenized in media through a 40 µm filter (Corning) under sterile conditions. Samples were centrifuged at 300 rcf for 5 min. The resulting pellet from the spleen sample and whole human blood were treated with RBC lysis buffer (7 min) and centrifuged at 300 rcf and the RBC lysis was repeated twice more until no RBC remained in the pellet. Samples were treated with human Fc-receptor blocking agent (TruStain FX, Biolegend) in flow buffer (HBSS, 0.1% BSA, 5 mM EDTA). For analysis, the following cocktail was used: hSig8 (PerCP/Cy5.5, clone 7C9, Biolegend cat. no. 347108, 1:200), hCD123 (PE/Cy7, clone 6H6, Biolegend cat. no. 306010, 1:200), hCD19 (PE, clone SJ25C1, Biolegend cat. no. 363004, 1:200), hCD3 (FITC, BD Pharmingen cat. no. 555332, 1:50), hCD14 (BV605, clone M5E2, BD Horizon cat. no. 564056, 1:200), hCD15 (BUV395, clone HI98, BD Horizon cat. no. 563872, 1:200), hCD56 (BV510, clone NCAM16.2, BD Horizon cat. no. 563041, 1:200). For human peripheral blood samples, we additionally used hHLA-DR (APC-Cy7, clone L243, Biolegend cat. no. 307618, 1:200). Pre-complexed Siglec-Fc/Strep-Tactin-AF647 (described above) was added to the antibody cocktail, added to cells, and allowed to stain at 4 °C for 45 min. The cells were then centrifuged at 300 rcf for 5 min and resuspended in flow buffer containing 1 µg/mL propidium iodide prior to analysis by flow cytometry.

**Flow cytometry**. Flow cytometry was performed on a BD LSRFortessa™ X-20 cell analyzer using BD FACSDiva Software Version 8.0.1 and data was processed using FlowJo LLC. Version 9.9.6 and Graphpad Prism 8. The live, single cell populations were gated out with two gates to obtain the median fluorescence intensity (MFI), which indicates binding of the Siglec-Fc-secondary complex to the cell surface. Control cells that were incubated with Strep-Tactin-AF647 (0.55 µg/mL, 0.57 pmol) was used to determine the baseline fluorescence. Number of cells (# cells) plotted indicates percentage of max (% max).

**Gel filtration of siglec-7-Fc/strep-tactin-AF647 pre-complex**. Purified Siglec-7-Fc was incubated with Strep-Tactin-AF647 as described above and the protein complex was injected into HiPrep 16/60 Sephacryl S-500 HR equilibrated with degassed Column Buffer (250 mM NaCl, 10 mM Tris-HCl, pH 7.5) at 4 °C. Protein fractions were collected using a flow rate of 0.5 mL/min and fractions were either stored at 4 °C or lyophilized and stored at −20 °C. AF647 fluorescence in fractions was determined by the fluorescence of the fractions (excitation: 650 nm; emission: 665 nm).

**Treatment of cells and tissues with neuraminidase**. K562 cells were incubated with Neuraminidase A at 37 °C for 30 min with shaking every 5 min. Cells were then added to a 96-well U-bottom plate and incubated with pre-complexed Strep-Tactin-AF647/Siglec, as described above.

**Formulation of CD33 ligand-targeted liposomes**. Commercially available lipids—DSPC and cholesterol—were suspended in chloroform and an appropriate volume of each lipid solution in chloroform was transferred into a glass test tube to reach the desired mol% (60.9% DSPC and 38% cholesterol) of each lipid. The solvent was removed under nitrogen gas to form the lipid mixtures. Once all visible chloroform was removed, 100 µL of dimethyl sulfoxide (DMSO) was added to the

test tube. CD33L-DSPE (compound 6, 1 mol%) and DSPE-PEG-A647 (0.1 mol%) in DMSO were then added to the lipid mixture. The samples were then placed at −80 °C until completely frozen and excess DMSO was removed via lyophilization overnight and then they dried liposomes were stored at −80 °C until they were extruded. Dried lipids were then allowed to warm to RT and were then hydrated with 1.0 mL of phosphate-buffered saline pH 7.4. The hydrated lipids were then sonicated in a cycle of 1 min on, 4–5 min off until all lipids were uniformly suspended. The lipids were then extruded with an 800 nm filter and then 100 nm filters. The average diameter of the liposomes was verified by dynamic light scattering (Malvern Panalytical Zetasizer Nano S) to be 110 ± 20 nm. Liposomes were stored at 4 °C.

**Immunofluorescence detection of Siglec-Fc binding to spleen.** Portions of human spleen were suspended in OCT and stored at −80 °C until ready for sectioning. Sections of human spleen tissue was sliced to 8 μm using a cryostat. The sections were placed on slides and soaked in PBS for 30 min. The slices were fixed with cold acetone for 10 min at 4 °C and then washed three times with PBS for 5 min each. A hydrophobic layer was made around the slice with a hydrophobic pen and the tissue was blocked in 5%/PBS for 1 h at RT. Each primary antibody was diluted (1:200) in 5% FBS/PBS, added to each section and incubated overnight at 4 °C in a humidity chamber. The sections were washed three times with PBS for 5 min each and the Siglec (40 μg/mL) preincubated with Strep-Tactin-AF647 was added to the section and incubated for 1 h at RT. The sections were then washed three times with PBS for 5 min each and then incubated with DAPI (1:100) for 25 min at RT. The sections were then mounted with antifade mounting media, a coverslip was added, and it was sealed with nail polish. The slides were stored at 4 °C until analysis.

**Immunofluorescence detection of siglec-Fc binding to breast cancer cell lines.** All cell lines were seeded onto coverslips at a density of 100,000 cells per coverslip. The following day, media was removed, and coverslips were washed three times with PBS. The cells were fixed with 4% paraformaldehyde in PBS for 15 min at room temperature. After, the cells were washed three times with PBS and then incubated with either Siglec-7-Fc WT or Siglec-7-Fc R124A that was preincubated with Strep-Tactin-AF647 as described above for 1 h at room temperature. The cells were then washed three times with PBS and incubated with DAPI for 10 min at RT. The cells were then washed with PBS once and mounted onto slides containing Pro-Long Gold Antifade mounting medium and stored at 4 °C until analysis.

**Microscopy for immunofluorescence of cells and tissues.** Breast cancer cell lines and spleen tissues were imaged using AxioCam digital microscope camera (Zeiss LSM-700, Germany) and processed using Zen (Blue Edition, Zeiss) Image Processing Software. Breast cancer cell lines were imaged at 20x magnification and spleen tissues were imaged at 63x magnification under oil immersion using Z-stacking every 0.2 μm for 5 μm and deconvoluted with Zeiss software.

**Immunohistochemistry detection of siglec-Fc binding to breast cancer tissues.** Human paraffin embedded breast cancer tissue microarrays (TMAs) were obtained from US Biomax Inc. (Rockville, MD, USA). Slides were dried overnight at 37 °C, deparaffinized twice in xylene, and rehydrated using a decreasing ethanol gradient followed by PBS. Heat-induced antigen retrieval was performed at 95 °C for 40 min in a 10 mmol/L citrate buffer (pH 6). After heating, slides were cooled down to room temperature (RT) and were washed in phosphate-buffered saline (PBS) solution. Endogenous peroxidase activity was quenched by submerging slides with BLOXALL® Endogenous Blocking saline (Vector Laboratories). Slides were then incubated for 20 min in blocking buffer, including 10% horse serum to prevent non-specific binding. Siglec-7-Fc WT, Siglec-7 R124A, Siglec-9-Fc WT, and Siglec-9 R120A proteins pre-complexed with Strep-Tactin®-HRP conjugate (IBA Lifesciences, USA) on ice for 20 min. TMAs were exposed to the conjugated solution for 1 h at RT. Excess reagents were removed by washing three times with PBS and visualized by DAB kit (Vector Laboratories). Finally, slides were counterstained with Harris hematoxylin (nuclear stain; blue signal), dehydrated, and mounted with Vecta mount medium. (Vector Laboratories). Siglec-7-Fc WT and Siglec-7-Fc R124A staining was visualized by chromogenic DAB stain (brown signal). Each TMA core was scanned by the confocal microscope slide scanner (Zeiss, Germany) at ×20 magnification.

**ESI mass spectrometry.** The affinities of α2-3 and α2-6 sialosides for the three Siglec fragments were monitored using the direct ESI-MS assay. In all cases, a reference protein ($P_{ref}$), cytochrome C, was added to the solution in order to correct mass spectra for any nonspecific binding that occurred during the ESI process[50]. For a 1:1 protein–ligand complex, the dissociation constant ($K_D$) can be calculated from the abundance (Ab) ratio ($R$) of the ligand bound (PL) to free protein (P) ions (Eq. (1)) measured by ESI-MS for solutions of known initial concentration of protein ($[P]_0$) and ligand ($[L]_0$), Eq. (2):

$$R = \frac{Ab(PL)}{Ab(P)} = \frac{[PL]_{eq}}{[P]_{eq}} \quad (1)$$

$$K_D = \frac{[L]_0}{R} - \frac{[P]_0}{R+1} \quad (2)$$

In principle, $K_D$ can be determined from measurements performed using a single concentration of P and L. However, to establish a reliable $K_D$ for a low affinity interaction it is generally preferable to utilize a titration approach, wherein $[P]_0$ is maintained constant and $[L]_0$ is varied. The affinity is determined by fitting Eq. 2 to a plot of fraction bound protein ($R/(R + 1)$) versus $[L]_0$:

$$\frac{R}{R+1} = \frac{[P]_0 + [L]_0 + K_D - \sqrt{([P]_0 + [L]_0 + K_D)^2 - 4[P]_0[L]_0}}{2[P]_0} \quad (3)$$

The reported errors correspond to 2 standard deviations (95% confidence interval). Direct ESI-MS measurements were performed in positive ion mode on a Synapt G2S quadrupole-ion mobility separation-time of flight (Q-IMS-TOF) mass spectrometer (Waters UK Ltd., Manchester, UK) and Q-Exactive Orbitrap mass spectrometer (Thermo Fisher Scientific). In both cases, nanoflow ESI (nanoESI) was used. The nanoESI tips, with ~5 μm outer diameters (o.d.), were produced in-house from borosilicate capillaries (1.0 mm o.d., 0.68 mm i.d.) using a P-1000 micropipette puller (Sutter Instruments, Novato, CA). A voltage of ~1 kV was applied to a platinum wire inserted into the back end of the nanoESI tip and in contact with the sample solution. For the measurements performed on the Synapt G2S instrument, the Source temperature was 60 °C and the Cone, Trap and Transfer voltages were 20 V, 3 V, and 1 V, respectively. All other instrumental conditions were set to the default parameters. Data acquisition and processing were carried out using MassLynx (Waters, version 4.1). For the Q-Exactive Orbitrap instrument, the automatic gain control target, the maximum injection time, capillary temperature and S-lens RF level were set to $1 \times 10^6$, 100 ms, 150 °C and 100, respectively. The resolution was 17,500 at $m/z$ 200. Data acquisition and processing were carried out using Xcalibur (Thermo Fisher, version 4.1).

**Synthesis of trisaccharides**

*N-acetylneuraminosylα2-6-O-D-galactopyranosylβ1-4-O-D-glucopyranose (Neu5Acα2-6Lac, compound 1).* CMP-Sialic acid (26.92 mg, 1.5 eq, 0.043 mmol) and α2-6-Sialyltransferse, Pd2,6ST (0.071 mg/ml) was added to a solution of lactose (10 mg, 0.029 mmol) in Tris-HCl buffer (452 μL, 100 mM, 20 mM MgCl2, pH 8.8) (Supplementary Fig. 24). The reaction mixture was incubated at 37 °C for 5 h and monitored by thin layer chromatography (TLC) (ethanol:methanol:water:acetic acid, 4:2:1:0.1, v-v:v:v). The reaction was terminated by dilution with a 4-fold of 100% ethanol and put at −20 °C for one hour to precipitate the enzyme. Precipitated protein was centrifuged (3700 rcf, 10 min, 4 °C), the supernatant carefully decanted into a round bottom flask, and evaporated. The residue was resuspended in water and purified on a P2 column equilibrated in 20% NH4OH to yield a white compound (15 mg, 93%) with the expected mass (633.21 Da): 1H NMR (700 MHz, D2O): δ = 4.54 (d, J = 7.7 Hz, 1H), 4.49 (d, J = 7.7 Hz, 1H), 4.45 (dd, J = 2.8, 4.9 Hz, 1H), 4.11 (dd, J = 5.6 Hz, 1H) 3.96–4.15 (m, 3), 3.81–3.94 (m, 7H), 3.75 (d, J = 10.5 Hz, 1H), 3.61–3.71 (m, 7H), 3.6 (dd, J = 9.73, 8.1 Hz, 1H), 3.56 (dd, J = 9.73, 7.8 Hz, 1H), 3.35 (t, J = 8.5 Hz, 1H), 2.74 (dd, J = 4.74, 12.6 Hz, 1H), 2.06 (s, 3H), 1.77 (t, J = 12.6 Hz, 1H).

*N-acetylneuraminosylα2-3-O-D-galactopyranosylβ1-4-O-D-glucopyranose (Neu5Acα2-3Lac, compound 2).* Lactose (2 mg, 0.0058 mmol) and CMP-Sialic acid (7 mg, 2 eq, 0.011 mmol) were dissolved in Tris buffer solution (265 μL, 100 mM, 20 mM MgCl2, pH 7.4) and recombinant shrimp alkaline phosphatase (rSAP) (NEB) (3 μL) and α2-3 sialyltransferase, Cst6 (ST3) (0.21 mg/ml) were added (Supplementary Fig. 25). The reaction was placed in a shaking incubator (25 °C, 3 rcf, overnight) and was monitored by TLC (ethanol:methanol:water:acetic acid, 4:2:1:0.1, v-v:v:v). The solution was diluted with a 4-fold of 100% ethanol and put at −20 °C for one hour to precipitate the enzyme. Precipitated enzyme was centrifuged (3700 rcf, 10 min, 4 °C), the supernatant decanted into a round bottom flask, and evaporated. The residue was resuspended in water and purified on a P2 column equilibrated in 20% NH4OH. After evaporation and lyophilization, Neu5Acα2-3Lac was a white solid (3.1 mg, 94%) with the expected mass (633.21 Da): 1H NMR (700 MHz, D2O): δ = 4.66 (d, J = 8.4 Hz, 1H), 4.52 (d, J = 8.4 1H), 4.10 (dd, J = 2.8, 9.8 Hz, 1H), 3.61–3.96 (dd, 5.1 Hz, 1H), 3.84–3.88 (m, 4H), 3.78–3.80 (m, 9H), 3.58–3.60 (m, 5H), 3.28 (t, J = 8.4 Hz, 1H), 2.76 (dd, J = 4.9, 11.9 Hz, 1H), 2.11 (s, 3H), 1.79 (t, J = 11.9 Hz, 1H).

*N-acetylneuraminosylα2-6-O-D-galactopyranosylβ(1-4)-2-(acetylamino)-2-deoxy-D-glucopyranose (Neu5Acα2-6LacNAc, compound 3).* A solution of LacNAc (4.5 mg, 0.011 mmol), CMP-Neu5Ac (1.5 eq, 11 mg, 0.018 mmol) and recombinant α2-6 sialyltransferase, Pd2,6ST (0.071 mg/ml) in Tris-HCl buffer (205 μL, 100 mM, 20 mM MnCl2, pH 8.8) was incubated for 5 h at 37 °C and pH was continually adjusted to 8.8. The reaction was monitored by TLC (ethanol:methanol:water:acetic acid, 4:2:1:0.1, v-v:v) (Supplementary Fig. 26). The mixture was stopped by dilution with 4-fold of 100% ethanol and put at −20 °C for one hour to precipitate the enzyme. Precipitated protein was centrifuged (3700 rcf, 10 min, 4 °C), the supernatant decanted into a round bottom flask, and evaporated. The residue was resuspended in water and purified on a P2 column equilibrated in 20% NH4OH to

yield a white solid compound (6.5 mg, 85%) with the expected mass (674.24 Da): $^1$H-NMR (D$_2$O, 700 MHz) $\delta$ = 5.2 (d, $J$ = 2.1, 1H), 4.55 (d, $J$ = 2.1, 1H), 4.0 (dd, $J$ = 9.7, 5.6 Hz, 2H), 3.97–3.86 (m, 6H), 3.81–3.86 (m, 3H), 3.65–3.79 (m, 7H), 3.58–3.60 (m, 3H), 3.56 (t, $J$ = 7.9 Hz, 1H), 2.68 (dd, $J$ = 4.2, 12.7 Hz, 1H), 2.08 (s, 3H), 2.02 (s, 3H), 1.79 (t, $J$ = 11.9 Hz, 1H).

N-*acetylneuraminosylα2-3-O-D-galactopyranosylβ(1-4)-2-(acetylamino)-2-deoxy-D-glucopyranose (Neu5Acα2-3LacNAc, compound 4).* LacNAc (1.5 mg, 0.0039 mmol) and CMP-Sialic acid (4.81 mg, 2 eq, 0.0078 mmol) were dissolved in Tris buffer solution (200 μL, 100 mM, 20 mM MgCl$_2$, pH 7.4). Recombinant shrimp alkaline phosphatase (rSAP) (NEB) (3 μL) and α2-3-sialyltransferase, Cst6 (ST3) (0.21 mg/ml) were added and the reaction was placed in a shaking incubator (25 °C, 3 rcf, overnight) and was monitored by TLC (ethanol:methanol:water:acetic acid, 4:2:1:0.1, v:v:v:v) (Supplementary Fig. 27). The reaction was stopped by dilution with a 4-fold of 100% ethanol and put at −20 °C for 1 h to precipitate the enzyme. Precipitated protein was centrifuged (3700 rcf, 10 min, 4 °C), the super-natant carefully decanted into a round bottom flask, and evaporated. The residue was resuspended in water and purified on a P2 column equilibrated in 20% NH$_4$OH to yield a white solid compound (2 mg, 83%) with the expected mass (674.24 Da): $^1$H-NMR (D$_2$O, 700 MHz) $\delta$ = 4.59 (d, $J$ = 8.4 Hz, 1H), 4.43 (d, $J$ = 7.7, 1H), 4.15 (dd, $J$ = 3.5, 10.5 Hz, 1H), 3.81–3.92 (m, 6H), 3.68–3.78 (m, 6H), 3.57–3.66 (m, 5H), 3.02 (t, $J$ = 7.7 Hz, 1H), 2.75 (dd, $J$ = 4.2, 11.9 Hz, 1H), 2.06 (s, 3H), 1.89 (s, 3H), 1.8 (t, $J$ = 11.9 Hz, 1H).

*Synthesis of CD33L-DSPE (compound 6).* The coupling reaction was performed by dissolving CD33L-NH$_2$, compound 5, (3.1 mg, 3.18 μmol, 1.25 equiv) anhydrous dimethylformamide (DMF) (1.6 mL) in a dried vial. To this solution was added NHS-activated-DSPE (2.5 mg, 2.54 μmol, 1 equiv) and Et$_3$N (1.5 euiv.) in anhydrous DMF (2.8 mL) at RT (Supplementary Fig. 28). The reaction mixture was allowed to stir at RT for 18 h. The solvent was removed under reduced pressure and the remaining crude product was loaded to Sephadex G-100 gel filtration column using H$_2$O, and the crude product was purified using H$_2$O as an eluent to afford CD33L-DSPE conjugate (6) as white solid after lyophilization of fractions having the desired product. The formation of the desired conjugate was confirmed by analysis of $^1$H NMR and high resolution mass spectrometry (Supplementary Fig. 23).

*Yield.* (4.25 mg, 90%), coupling efficiency 89%. Coupling efficiency of compound 5 with the DSPE-PEG scaffold was determined by $^1$H NMR integration. $^1$H NMR (700 MHz, DMSO-D6): $\delta$ 7.70 (s, 1H), 7.48 (s, 2H), 5.06-5.04 (broad s, 1H), 5.02–4.99 (m, 2H), 4.62–4.58 (m, 1H), 4.29–4.25 (m, 2H), 4.21–4.18 (m, 1H), 4.07 (dd, $J$ = 7.0, 11.9 Hz, 1H), 3.74-3.73 (m, 5H), 3.68-3.65 (m, 2H), 3.58–3.52 (m, 2H), 3.24-3.22 (m, 3H), 3.17 (broad s, 2H), 3.10 (t, $J$ = 7.7 Hz, 1H), 3.05 (t, $J$ = 6.3 Hz, 1H), 2.39 (broad s, 1H), 2.26–2.23 (m, 4H), 2.17 (s, 6H), 2.06 (t, $J$ = 7.7 Hz, 1H), 2.02 (t, $J$ = 6.3 Hz, 1H), 2.06–2.02 (m, 4H), 1.92 (broad s, 2H), 1.73–1.69 (m, 3H), 1.66–1.64 (m, 1H), 1.48–1,47 (m, 4H), 1.34 (t, $J$ = 9.8 Hz, 4H), 1.22 (broad s, 64H), 0.84 (s, 6H).

**Reporting summary**. Further information on research design is available in the Nature Research Reporting Summary linked to this article.

## Data availability

The authors declare that all data supporting the findings of this study are available within the paper and its Supplementary information files. Please contact the corresponding author (M.S.M.) for access of raw data, which is stored electronically, and will be made available upon request. The coding sequence of Neuraminidase-A and Neuraminidase-S can be found in the GenBank; accession number: AY934539.2 for Neu-A and ABJ55283 for Neu-S. Source data are provided with this paper.

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

## Acknowledgements

We thank the UAlberta flow core facility with assistance in cell sorting, and the NMR and mass spectrometry facilities within the UAlberta Department of Chemistry for assistance with analytical services. We thank Dr. Christoph Rademacher for advice on coupling glycans to NHS-DSPE. M.S.M. thanks NSERC, Canada Research Chairs, Canadian Glycomics Network, and the Alberta Glycomics Centre for funding. E.R. thanks Alberta Innovates for a graduate student scholarship. C.L. thanks GlycoNet for an undergraduate summer student scholarship. We thank the GlycoNet Synthetic core for assistance in preparing the high affinity CD33 ligand. We thank Jean Pearcey for coordinating acquition of the spleen samples, and Jeremy Jerasi and Dr. Maya Shmulevitz for culturing and providing the breast cancer cell lines. We thank Dr. Warren Wakarchuck for providing the plasmid encoding the galactosyltransferase, Ruixiang Zhen for assistance with expression and purification of the sialyltransferases, and Melissa Gray and Dr. Carolyn Bertozzi for providing a detailed protocol for FGE expression and use.

## Author contributions

E.R. and M.S.M. designed the experiments and wrote the paper. J.J., C.L., and L.S. assisted in the cloning, expression, and probing of cells with Siglec-Fcs by flow cytometry. H.P., E.K., L.N., and J.K. designed and performed the mass spectrometry studies. S.So. and K.W. performed the IHC staining of breast cancer tissues. V.A., E.R., and S.Sa. performed the IF staining of human spleen. F.M. carried out the chemo-enzymatic synthesis of the trisaccharides, while GD conjugated the high affinity CD33 ligand to lipid and worked with ES to prepare and characterize liposomes. CDS generated the CRISPR KO cells. L.J.W. provided the human spleen. J.P.J. provided the His$_6$-tagged TEV.

## Competing interests

The authors declare no competing interests.
