## [Peer Review File · Nature Communications]

REVIEWER COMMENTS

Reviewer #1 (Remarks to the Author):

This manuscript by Emily Rodrigues et al. discusses the interaction of sialic acid-binding immunoglobulin-type lectins (Siglecs) with different carbohydrate ligands. Siglecs are immunomodulatory receptors expressed on the surface of certain cell types as transmembrane proteins. The physiological importance of these interactions has been getting a growing interest in the scientific community due to their connection to cancer, pathogenic infection, or as mentioned here: Alzheimer's disease.

The experimental work presented in this manuscript is meticulous, and there is no doubt that approaching this topic from different perspectives at different biological organizational levels (protein solution, 2D cell lines, tissues – both healthy and tumor-originated) makes the paper worthy of being published in a prestigious journal. However, I have a couple of concerns with the manuscript in its current form, some are minor, others are serious. If these issues are properly addressed, the manuscript may indeed become suitable for Nature Communications.

Language issues:

1) The manuscript contains a couple of grammatical, typographical, and stylistic errors that need to be fixed. Certain parts of the manuscript are very difficult to understand and even though the experimental work is outstanding, the manuscript itself is not of the quality required for publication in Nat. Comm. Here a couple of examples:

- Abstract: "A growing connection between Siglecs and human disease" This is a typical example of imprecise language, which is found in different parts of the manuscript. The connections have always been there, but our knowledge of them is what's emerging.

- There are a couple of vague expressions in the abstract ("many unique patterns", "additional features") that are not defined. Moreover, there imprecise statements in the methods section such as "appropriate volume", "approximately 100 μL ", etc.

- In the Methods section, the abbreviation “RT” is not defined. I would suggest to write the whole expression instead of using abbreviations, as this will make the text easier to read.

These are just a few examples and I strongly encourage the authors put some time into the general revision of the language.

2) In some cases, the small size of individual panels in the figures makes it difficult for the reader to find the tiny differences. In figure 3, the rough edge of the mutated Fc is not well visible, and the molecules illustrated in a and b appear the same. Also the axes of some figures (especially the x-axes of the cytometric panels) are difficult to understand both because of the small ticks and the ambiguous axes title (“Siglec-XX binding” instead of the measured quantities). The authors should make sure that the figures are better comprehensible.

3) The motivation of the manuscript is not clearly described in the abstract, introduction, and the conclusion.

Scientific issues:

4) The caption of Fig. 3 a, b is not consistent with the plot labels in the figure. While the caption says that the WT is marked with a black line while the R120A mutant is marked with a red line, the colors in the panels seem to depend on whether the Fc was or was not blocked.

5) On page 8, the authors state “With V1, significant binding is observed to neutrophils with only a small reduction in the R120A mutant. When Fc blocking antibodies were used, binding of both the WT and R120A mutant decreased...” Based on the labels in Fig. 3a, the mutation in the Fc caused at least as much decrease, if not more, than the blocking of Fc. Moreover, it is not defined anywhere in the text what the curve with the gray filling is (I suppose, it is the control).

6) There is no reasoning in the introduction/discussion/methods, why the authors used the chosen cell lines. While the number of different cell lines is very promising, a short discussion should be added to show that the choice of cell lines was not arbitrary.

7) On page 10, the authors write “Noteworthy binding was observed for Siglec-1 to monocytes, CD22 to B-cells, ...” however, in the case of Siglec-1, basophils or pDC cells show a stronger binding based on Fig. 3d. What is the reason the authors highlighted only monocytes, a cell type that has a lower affinity to bind than some other cells?

8) On page 10, the authors write “As highlighted for binding of Siglec-1 to B-cells (Fig. 3e), [...] significantly enhanced binding was observed in the pre-complexed conditions”. However, Fig. 3e is supposedly data collected from monocytes, not B-cells. Also, the phrase “significantly enhanced” could maybe be used for Fig. 3g, where the increase in MFI is about an order of magnitude, but in the case of Fig. 3e, it only seems to be enhanced approximately 4-fold. It would be more fortunate to use more exact expressions throughout the text, e.g., “x-fold” than using words like “significantly”.

9) On page 10, it is stated “On the tissues, we observe staining of Siglec-7-Fc on all cell types (Fig. 3h),...” however, no discussion can be found in the text as to what cell types the authors refer to in the spleen tissue sample. The IF microscopy images are quite dark, and the structure of the section is difficult to see. If the authors have the data, bright-field microscopy images should be provided at least in the Supplementary Information with the same sample position.

10) Page 11: “Strong staining of Siglec-Fc...” Not all readers are familiar with immunohistochemistry. A short discussion should be added either in-text or in the figure caption as to what they should look for when it comes to “strong staining”.

11) In Supplementary Fig. 14, the tissue sections should be labelled by their type, just like in Fig. 4 c,d.

12) Page 11: “To directly detect Siglec-glycan interactions by ESI-MS, a key obstacle was their large size and heterogeneity, owing to many glycoforms”. In Fig. 5b, the authors present the ESI-MS spectrum of the whole chimera complex, which proves that their large size was not an obstacle at all. If a spectrum with higher S/N is desired, a longer acquisition time can be used. Based on the spectra, the heterogeneity due to glycoforms seems to be the major problem.

13) Fig. 5e: What do the dashed lines mark? If they mark the peak positions in the bottom panel, their position should be readjusted.

14) Page 14: “Titrating the concentration of ligand enabled the KD values to be determined (Fig. 5f, Supplementary Fig. 17-19).” The figures only show single concentrations; Similar to Supplementary Fig. 20, the titration curves should be shown in the Supplementary Information. Moreover, here (Supplementary Fig. 17-19), the choice of Siglec and ligand concentrations seem to be very arbitrary,

covering a wide range among the figures. Why did the authors choose different concentrations in the different figures that are meant to be comparable? How does it affect their comparability?

15) Page 28: “The size of the liposomes was then verified by dynamic light scattering [...] to be approximately 110 nm”. Given the nature of DLS results, the authors should insert here the average size with the standard deviation instead of using the word “approximately”.

16) On page 31, the authors mention that the source temperature was 60 °C. It has been shown multiple times in the past (e.g., Timothy D Veenstra et al., *Journal of the American Society for Mass Spectrometry*, 1998, 9, 580-584) that an increased source temperature can promote the dissociation of non-covalently bound complexes. This is especially true to complexes that have a weak binding affinity. While the authors discuss that the so-far used experimental approaches might not reflect real KD values, their own experimental conditions might as well result in weaker experimental binding affinities. Therefore, the authors should provide proof (preferably in the form of data collected with source temperatures below 40 °C) that the KD values they measured were not influenced by a high source temperature.

Reviewer #2 (Remarks to the Author):

Summary:

The paper by Rodrigues et al. describes a new library of recombinant human Siglecs, a family of glycan recognition proteins with immune checkpoint-like properties. The author engineered the human IgG1-Fc region (common scaffold) to improve the versatility of the protein library, and used these reagents to profile the expression patterns of Siglec ligands on human normal tissues (spleen and peripheral blood) and tumor (breast cancer). They also demonstrated that these recombinant Siglecs are useful for the determination of their glycan binding specificity and precise affinity constants for pure synthetic glycans.

Comments:

The library of recombinant human Siglecs prepared by the authors is by far the most comprehensive and best-characterized reagent set focusing on Siglecs. The authors demonstrated the versatility and

usefulness of the library for a wide array of applications. These reagents will be invaluable resource for those who are interested in the biological functions of Siglecs.

I have a question and some minor comments.

1. Question

In Figure 1b, the sizes of CD33-Fc and Siglec-15-Fc appear to be much smaller (by ~30 kDa?) than those of other Siglec-Fcs. Although some size difference is expected (as CD33-Fc and Siglec-15-Fc contain only two Ig-like domains of these Siglecs, whereas other Siglec-Fcs incorporate three Ig-like domains), the size difference on SDS-PAGE appear to be too large to be accounted for by one Ig-like domain difference. Do the authors have any explanation?

2. Comments

(1) Page 8, line 7-8: Sample descriptions (293 and K562) and figure numbers (Supplementary Figure 7a and 7b) appear to be inverted.

(2) Page 8, line 10-11: The “Fc blocking antibodies” the author used (Human TruStain FcX, BioLegend) appears to “contain specialized human IgG” (although its specificity or formulation is not disclosed) and “not recommended to be used for staining human IgG”, according to the product datasheet. Thus, the phrase “Fc blocking antibody” may not be precise. It may better be described as “Fc receptor blocking reagent”.

(3) Page 38, ref 43: Only the initials of the authors are shown.

(4) Supplementary Figure 5: Figure legend appears to be somewhat disorganized. Panel c likely represents fraction-by-fraction K562 binding signal before lyophilization, and panel d likely represents that after lyophilization, but these are not obvious.

Reviewer #3 (Remarks to the Author):

Rodrigues et al provide a very thorough and powerful set of tools for identification of Siglec ligands. The tools are likely to gain broad use and enhance discovery in the field of functional glycosciences. The set consists of 14 sialic-acid binding Siglecs expressed as soluble chimeras and their non-binding site mutants. Each contains up to three Siglec extracellular domains attached to well-established tags for biological research. While no one tag is novel, together - and as a set - they provide powerful functionality for different kinds of biological and biochemical experiments as established by

examples in the manuscript. These include flow cytometry, immunocytochemistry, immunohistochemistry, and mass spectrometric binding affinity. One can envision this versatile and complete set of human Siglec tools adding significantly to the capabilities and discoveries in this area of research.

Minor corrections and changes are recommended:

p. 2: “systemically” meant to be “systematically”?

p.2: State that “CD33” is Siglec-3 in the abstract.

p.4: The schematic does not show the His-6-tag as “C-terminal”

p.7: There are two peaks in the sizing column of Siglec-7 complex. This should be acknowledged.

p.8: Something appears to be missing in the sentence: “Fc[gamma]R are abundant on immune cells and binding of Siglec-Fc proteins to cells.”

p.9: The reduction of V1 binding (Fig. 3a) appears to be ~3-fold and that of V2 (Fig. 3b) ~2-fold, yet the former is noted as “only a small reduction” and the latter as “significantly decreased”.

p.9: “...by Fc blocking antibodies (Fig. 3C)” refers to a panel with no Fc blocking antibodies used.

p. 10: Fig. 3 legend uses the term “mutant” which should be noted as “arginine mutant” to distinguish it from the Fc mutations.

p. 17: “all Siglecs” does not include Siglec-12. The text should somewhere acknowledge that Siglec-12 was not used in most experiments.

p. 24: “Neuraminidase-B” meant to be “Neuraminidase-S”?

p. 38: Reference 43 is missing author last names.

Reviewer #1 (Remarks to the Author):

Language issues:

1) *The manuscript contains a couple of grammatical, typographical, and stylistic errors that need to be fixed. Certain parts of the manuscript are very difficult to understand and even though the experimental work is outstanding, the manuscript itself is not of the quality required for publication in Nat. Comm. Here a couple of examples:*

- Abstract: "A growing connection between Siglecs and human disease" This is a typical example of imprecise language, which is found in different parts of the manuscript. The connections have always been there, but our knowledge of them is what's emerging.

Response: Thank you for pointing this out. To address this, we have modified the abstract to more simply state: "The connections between Siglecs and human disease motivates improved methods to detect Siglec ligands".

- There are a couple of vague expressions in the abstract ("many unique patterns", "additional features") that are not defined. Moreover, there imprecise statements in the methods section such as "appropriate volume", "approximately 100 μ L", etc.

Response: These are excellent observations and we thank the reviewer for bringing them up. For the first point ("many unique patterns" and "additional features") we are limited by a short abstract (150 words), which does not allow for these descriptors to be expanded on. With regards to the imprecise statements, we agree with the reviewer. Numerous sentences have been changed to report the actual value that was measured.

- In the Methods section, the abbreviation "RT" is not defined. I would suggest to write the whole expression instead of using abbreviations, as this will make the text easier to read. These are just a few examples and I strongly encourage the authors put some time into the general revision of the language.

Response: We thank the review for pointing out this lack of definitions. RT has been defined as well as several other abbreviations defined.

2) *In some cases, the small size of individual panels in the figures makes it difficult for the reader the find the tiny differences. In figure 3, the rough edge of the mutated Fc is not well visible, and the molecules illustrated in a and b appear the same. Also the axes of some figures (especially the x-axes of the cytometric panels) are difficult to understand both because of the small ticks and the ambiguous axes title ("Siglec-XX binding" instead of the measured quantities). The authors should make sure that the figures are better comprehensible.*

Response: In the revised manuscript, we have imported higher resolution files into the manuscript, which should help address some of these concerns. For the axis titles, we have used descriptors of what they are measuring, rather than the fluorophore, because we feel this will help the reviewer easily interpret the data. However, we see the point of the reviewer and to address this we have pointed out the fluorophore being measured (AF647 in most cases) in the figure legend.

3) *The motivation of the manuscript is not clearly described in the abstract, introduction, and the conclusion.*

Response: Thank you for the suggestion. As discussed above in the 1st point, the abstract already states, “The connection between Siglecs and human disease motivates improved methods to detect Siglec ligands.” However, we found a few places in the introduction and discussion to improve our communication of the motivation. Specifically, in the introduction, we now write, “we were motivated to improving this scaffold, through eliminating the aforementioned drawbacks, would be highly desirable” and in the discussion, “Motivated to better understanding Siglec ligands as self-associated molecular patterns, we have developed a new set of soluble Siglecs to probe Siglec-glycan interactions.”

Scientific issues:

4) *The caption of Fig. 3 a, b is not consistent with the plot labels in the figure. While the caption says that the WT is marked with a black line while the R120A mutant is marked with a red line, the colors in the panels seem to depend on whether the Fc was or was not blocked.*

Response: Thank you for pointing out this error. The figure legend has been modified to correctly represent the conditions.

5) *On page 8, the authors state “With V1, significant binding is observed to neutrophils with only a small reduction in the R120A mutant. When Fc blocking antibodies were used, binding of both the WT and R120A mutant decreased...” Based on the labels in Fig. 3a, the mutation in the Fc caused at least as much decrease, if not more, than the blocking of Fc. Moreover, it is not defined anywhere in the text what the curve with the gray filling is (I suppose, it is the control).*

Response: This is correct – the mutations to block Fc-engagement have a larger effect than Fc blocking. This is consistent with Fc blocking not being 100% efficient. We feel that adding text to describe this would distract from the main point, which is that the V2 versions of our constructs greatly enhance sialic acid-dependent binding to cells. The grey-filled curves are the controls with Strep-Tactin-AF647 alone. Thank you for pointing out that this was missing from the figure legend – it has now been added.

6) *There is no reasoning in the introduction/discussion/methods, why the authors used the chosen cell lines. While the number of different cell lines is very promising, a short discussion should be added to show that the choice of cell lines was not arbitrary.*

Response: For Figure 2, we selected a diverse array of cell lines that are representative of different origins (e.g. kidney blood cells, lung, and intestinal). In the results, we have extended a sentence to include a description of this, “We next examined binding of pre-complexed Siglec-Fc chimeras with Strep-Tactin to a variety of cell lines that are representative of different origins”. In Figure 4, the cell lines are all breast cancer cell lines, which is well described.

7) *On page 10, the authors write “Noteworthy binding was observed for Siglec-1 to monocytes, CD22 to B-cells, ...” however, in the case of Siglec-1, basophils or pDC cells show a stronger binding based on Fig. 3d. What is the reason the authors highlighted only monocytes, a cell type that has a lower affinity to bind than some other cells?*

Response: We thank the reviewer for bringing up this point. The description of the data in the text has been improved to simply report what cell type each Siglec bound best.

8) On page 10, the authors write “As highlighted for binding of Siglec-1 to B-cells (Fig. 3e), [...] significantly enhanced binding was observed in the pre-complexed conditions”. However, Fig. 3e is supposedly data collected from monocytes, not B-cells. Also, the phrase “significantly enhanced” could maybe be used for Fig. 3g, where the increase in MFI is about an order of magnitude, but in the case of Fig. 3e, it only seems to be enhanced approximately 4-fold. It would be more fortunate to use more exact expressions throughout the text, e.g., “x-fold” than using words like “significantly”.

Response: We thank the reviewer for pointing out this discrepancy. The figure and the text have been changed to be consistent. With regards to the how the data is described, we agree with the reviewer that reporting the fold change would be appropriate here. Therefore, we have modified the sentence to: “As highlighted for binding of Siglec-1 to monocytes (**Fig. 3e**) there was a 3-fold increase in binding in the pre-complexed conditions, as well as a 5-fold increase for Siglec-7 to eosinophils (**Fig. 3f**), and a 4-fold increase for Siglec-9 to basophils (**Fig. 3g**).”

9) On page 10, it is stated “On the tissues, we observe staining of Siglec-7-Fc on all cell types (Fig. 3h),...” however, no discussion can be found in the text as to what cell types the authors refer to in the spleen tissue sample. The IF microscopy images are quite dark, and the structure of the section is difficult to see. If the authors have the data, bright-field microscopy images should be provided at least in the Supplementary Information with the same sample position.

Response: This is a good point the reviewer brings up. We performed co-staining with numerous cells types in these experiments, but what we found is that by IF staining, which is necessarily less quantitative than flow cytometry, Siglec-7 did not show any particular preference but was binding uniformly to all cell types in a sialic acid dependent manner, are shown in the Figure. These results are consistent with our flow cytometry that Siglec-7 bound to many cell types found in the spleen. To clarify this point, we have modified the last sentence in this section to read, “The ability of Siglec-7-Fc to bind many cell types is consistent with broad staining of Siglec-7-Fc on splenocytes from the same spleen by flow cytometry (**Supplementary Fig. 12**).”

10) Page 11: “Strong staining of Siglec-Fc...” Not all readers are familiar with immunohistochemistry. A short discussion should be added either in-text or in the figure caption as to what they should look for when it comes to “strong staining”.

Response: Thank you for the suggestion. We have added a description at the end of the figure legend for the non-expert, “Positive signal is represented by brown coloration from chromogenic staining. Darker staining as represented by increased brown coloration indicates Siglec-7-Fc binding (c) as compared to control (d), while blue staining represents cell nuclei (hemotaxilin stain).” More details have also been added to the methods.

11) In Supplementary Fig. 14, the tissue sections should be labelled by their type, just like in Fig. 4 c,d.

Response: Thank you for pointing this out. This addition has been made.

12) Page 11: “To directly detect Siglec-glycan interactions by ESI-MS, a key obstacle was their large size and heterogeneity, owing to many glycoforms”. In Fig. 5b, the authors present the ESI-MS spectrum of the whole chimera complex, which proves that their large size was not an obstacle at all. If a spectrum with higher S/N is desired, a longer acquisition time can be used. Based on the spectra, the heterogeneity due to glycoforms seems to be the major problem.

Response: We agree with the reviewer that the sizes of the Siglecs do not prevent their direct analysis by ESI-MS. However, the heterogeneity arising from glycosylation, combined with the higher charge states prevents reliable ligand binding measurements from being performed by ESI-MS. As examples, consider the upper panels of Supplementary Figure 16 for CD22 and CD33. The mass spectra of the entire chimeras are extremely broad, owing to the multiple glycoforms and the different glycoforms are not baseline resolved. As such, quantitative measurements of a glycan complex would be extremely challenging or impossible. Creating the sharp peaks, by removing the glycans and Fc domain, allows the glycan complex to be clearly identified and accurately quantified.

13) *Fig. 5e: What do the dashed lines mark? If they mark the peak positions in the bottom panel, their position should be readjusted.*

Response: Thank you for pointing out this ambiguity. The dashed lines are there to help orient the reader's eye that one peak in the top spectra corresponds to the same peak in the bottom spectra. A description of what these dashed lines represent has been added to the figure legend. We agree with the reviewer of their suboptimal positioning, and have improved in their positioning in the revised manuscript.

14) *Page 14: "Titrating the concentration of ligand enabled the KD values to be determined (Fig. 5f, Supplementary Fig. 17-19)." The figures only show single concentrations; Similar to Supplementary Fig. 20, the titration curves should be shown in the Supplementary Information. Moreover, here (Supplementary Fig. 17-19), the choice of Siglec and ligand concentrations seem to be very arbitrary, covering a wide range among the figures. Why did the authors choose different concentrations in the different figures that are meant to be comparable? How does it affect their comparability?*

Response: To address this concern we have added the titration curves to Supplementary Figures 17-19.

15) *Page 28: "The size of the liposomes was then verified by dynamic light scattering [...] to be approximately 110 nm". Given the nature of DLS results, the authors should insert here the average size with the standard deviation instead of using the word "approximately".*

Response: This was an oversight on our part, and we thank the reviewer for bringing it up. Indeed, the reported values do indeed represent the average diameter and a standard deviation. We have modified the description around this in the methods to read, "The average diameter of the liposomes was verified by dynamic light scattering (Malvern Panalytical Zetasizer Nano S) to be 110 +/- 20 nm."

16) *On page 31, the authors mention that the source temperature was 60 °C. It has been shown multiple times in the past (e.g., Timothy D Veenstra et al., Journal of the American Society for Mass Spectrometry, 1998, 9, 580-584) that an increased source temperature can promote the dissociation of non-covalently bound complexes. This is especially true to complexes that have a weak binding affinity. While the authors discuss that the so-far used experimental approaches might not reflect real KD values, their own experimental conditions might as well result in weaker experimental binding affinities. Therefore, the authors should provide proof (preferably in the form of data collected with source temperatures below 40 °C) that the KD values they measured were not influenced by a high source temperature.*

Response: We agree with the reviewer that in-source dissociation represents a potential source of error for ESI-MS protein-ligand binding measurements. However, K_D is not a good indicator of the susceptibility of complexes to undergo such dissociation. Rather, for protein-glycan interactions, the size of the ligand and the number of hydroxyl groups available to form stabilizing intermolecular H-bonds (in the gas phase) is the most reliable predictor. It has been shown previously that the gas-phase kinetic stabilities of protein complexes with trisaccharide (and larger) ligands, which were used in the current study, are sufficient to allow them to be preserved under gentle (low excitation) source conditions on multiple MS platforms (e.g. Wang *Anal. Chem.* **2003**, 75, 4945-4955; Rademacher *JACS* **2007**, 129, 10489-10502; Yao *JASMS* **2015**, 26, 98-106; Km Shams-Ud-Doha et al. *Anal. Chem.* **2017**, 89, 4914-4921). In the case of the Synapt G2S instrument, the use of a 60 °C Source temperature has been shown to give reliable values for such measurements (Han, L. *Glycobiology*, **2018**, 28, 488-498). However, as we have not previously reported on protein-glycan affinity measurements using the Q-Exactive Orbitrap, we performed a series of binding measurements on the Siglec-1 fragment interaction with Neu5Ac α 2-6Lactose using a range of capillary temperatures (results presented below for review only). In this experiment, a mixture of 3.6 μ M Siglec-1 fragment and 80 μ M Neu5Ac α 2-6Lactose were analyzed and the source temperature was varied between 100 and 200 °C. As shown in the data below, the measured K_D values are the same (within experimental error) over this temperature range. These results indicate that the contribution of in-source dissociation to the K_D measurements is negligible.

Reviewer #2 (Remarks to the Author):

The library of recombinant human Siglecs prepared by the authors is by far the most comprehensive and best-characterized reagent set focusing on Siglecs. The authors demonstrated the versatility and usefulness of the library for a wide array of applications. These reagents will be invaluable resource for those who are interested in the biological functions of Siglecs.

Response: Thank you for the positive feedback.

I have a question and some minor comments.

1. Question

In Figure 1b, the sizes of CD33-Fc and Siglec-15-Fc appear to be much smaller (by ~30 kDa?) than those of other Siglec-Fcs. Although some size difference is expected (as CD33-Fc and Siglec-15-Fc contain only two Ig-like domains of these Siglecs, whereas other Siglec-Fcs incorporate three Ig-like domains), the size difference on SDS-PAGE appear to be too large to be accounted for by one Ig-like domain difference. Do the authors have any explanation?

Response: We thank the reviewer for point this out. The discrepancy is due to mislabelling the molecular weight marker. Once corrected, the difference in MW is only 15-20 kDa, which is consistent with the size of the missing Ig-like domain plus any associated glycosylation.

2. Comments

(1) Page 8, line 7-8: Sample descriptions (293 and K562) and figure numbers (Supplementary Figure 7a and 7b) appear to be inverted.

Response: Thank you for pointing out this discrepancy. The text has been adjusted to reflect the order it is presented in Supplementary Figure 7a and 7b.

(2) Page 8, line 10-11: The “Fc blocking antibodies” the author used (Human TruStain FcX, BioLegend) appears to “contain specialized human IgG” (although its specificity or formulation is not disclosed) and “not recommended to be used for staining human IgG”, according to the product datasheet. Thus, the phrase “Fc blocking antibody” may not be precise. It may better be described as “Fc receptor blocking reagent”.

Response: This is an excellent point and thank the reviewer for bringing it up. We agree that describing it as an agent is better. Therefore, we have changed the description from antibody to agent throughout.

(3) Page 38, ref 43: Only the initials of the authors are shown.

Response: Thank you for pointing out this error. This issue has been resolved.

(4) Supplementary Figure 5: Figure legend appears to be somewhat disorganized. Panel c likely represents fraction-by-fraction K562 binding signal before lyophilization, and panel d likely represents that after lyophilization, but these are not obvious.

Response: Thank you for point this out. The figure legend has been improved to be describe the individual panels more accurately.

Reviewer #3 (Remarks to the Author):

Rodrigues et al provide a very thorough and powerful set of tools for identification of Siglec ligands. The tools are likely to gain broad use and enhance discovery in the field of functional glycosciences. The set consists of 14 sialic-acid binding Siglecs expressed as soluble chimeras and their non-binding site mutants. Each contains up to three Siglec extracellular domains attached to well-established tags for biological research. While no one tag is novel, together - and as a set - they provide powerful functionality for different kinds of biological and biochemical experiments as established by examples in the manuscript. These include flow cytometry, immunocytochemistry, immunohistochemistry, and mass spectrometric binding affinity. One can envision this versatile and complete set of human Siglec tools adding significantly to the capabilities and discoveries in this area of research.

Response: Thank you for the positive feedback.

Minor corrections and changes are recommended:

p. 2: “systemically” meant to be “systematically”?

Response: Thank you for pointing out this error. This error, as well several others, have been fixed.

p.2: State that “CD33” is Siglec-3 in the abstract.

Response: Thank you for the suggestion. This clarification has been added.

p.4: The schematic does not show the His-6-tag as “C-terminal”

Response: Thank you for pointing out this discrepancy. Figure 1 is correct and so we have changed the description in the text to, “a His₆-tag, for purification, directly C-terminal to the Fc domain”.

p.7: There are two peaks in the sizing column of Siglec-7 complex. This should be acknowledged.

Response: This is true. This peak likely is a very small amount of aggregated protein. However, we note that we never see any bands for this peak on a gel. To acknowledge this peak, we have added the following sentence: “A smaller peak also appeared in the void volume, which may represent a small amount of aggregated protein.”

p.8: Something appears to be missing in the sentence: “Fc[gamma]R are abundant on immune cells and binding of Siglec-Fc proteins to cells.”

Response: Thank you for pointing out this omission. The sentence has been extended to, “Fc γ R are abundant on immune cells and binding of Siglec-Fc proteins to cells may take place through Fc γ R instead of sialic acid.”

p.9: The reduction of V1 binding (Fig. 3a) appears to be ~3-fold and that of V2 (Fig. 3b) ~2-fold, yet the former is noted as “only a small reduction” and the latter as “significantly decreased”.

Response: Thank you for pointing this out, we agree that better wording could have been used to accurately describe the data. Accordingly, we have changed the description of the R120A to point

out how much binding there is relative to background. These sentences now read: “With V1, significant binding is observed to neutrophils with a reduction in the R120A mutant to levels that are still significantly above background (**Fig. 3a**)” and “The V2 of CD22-Fc showed lower binding, the reduced R120A mutant binding was much closer to background, and was unaffected by Fc blocking agent (**Fig. 3b**)”

p.9: “...by Fc blocking antibodies (Fig. 3C)” refers to a panel with no Fc blocking antibodies used.

Response: Thank you for pointing this out. We have the data to support the lack of effect from Fc blocking but it was not included as we did not feel it was needed since the R120A mutant is already background. Therefore, we have removed the part, “and was not affected by Fc blocking antibodies”.

p. 10: Fig. 3 legend uses the term “mutant” which should be noted as “arginine mutant” to distinguish it from the Fc mutations.

Response: Thank you for pointing out this oversight. We have added ‘arginine’ to the figure legend as suggested.

p. 17: “all Siglecs” does not include Siglec-12. The text should somewhere acknowledge that Siglec-12 was not used in most experiments.

Response: We thank the reviewer for pointing this out. First, we have now included all the data for Siglec-12 binding to cell lines in Figure 2 and have added the following sentence, “As Siglec-12 did not show binding to any of the cell lines, it was not used in subsequent experiments.”. Furthermore, on p. 17 of the discussion an addition has been made, “Consistent with this, all Siglecs – except Siglec-5/14 and 11/16, which share an identical V-set domain with their pair, as well as Siglec-12 that lacks sialic acid binding – showed a unique binding pattern to cells.”

p. 24: “Neuraminidase-B” meant to be “Neuraminidase-S”?

Response: Thank you for pointing out this error. Indeed, it was supposed to be ‘S’ instead of ‘B’. This change has been made.

p. 38: Reference 43 is missing author last names.

Response: This issue has been corrected.